# CONSTRAINT MATTERS: MULTI-MODAL REPRESENTATION FOR REDUCING MIXED-INTEGER LINEAR PROGRAMMING

Jiajun Li[1], Yixuan Li[1], Ran Hou[1], Yu Ding[1], Shisi Guan[2], Jiahui Duan[2], Xiongwei Han[2], Tao Zhong[2], Vincent Chau[1], Weiwei Wu[1], Zhiyuan Liu[1], and Wanyuan Wang[1, *]

[1]School of Computer Science and Engineering, Southeast University
[2]Huawei Noah's Ark Lab
[*]Corresponding author: Wanyuan Wang(wywang@seu.edu.cn)

## ABSTRACT

Model reduction, which aims to learn a simpler model of the original mixed integer linear programming (MILP), can solve large-scale MILP problems much faster. Most existing model reduction methods are based on variable reduction, which predicts a solution value for a subset of variables. From a dual perspective, constraint reduction that transforms a subset of inequality constraints into equalities can also reduce the complexity of MILP, but has been largely ignored. Therefore, this paper proposes a novel constraint-based model reduction approach for MILPs. Constraint-based MILP reduction has two challenges: 1) which inequality constraints are critical such that reducing them can accelerate MILP solving while preserving feasibility, and 2) how to predict these critical constraints efficiently. To identify critical constraints, we label the tight-constraints at the optimal solution as potential critical constraints and design an information theory-guided heuristic rule to select a subset of critical tight-constraints. Theoretical analyses indicate that our heuristic mechanism effectively identify the constraints most instrumental in reducing the solution space and uncertainty. To learn the critical tight-constraints, we propose a multi-modal representation that integrates information from both instance-level and abstract-level MILP formulations. The experimental results show that, compared to the state-of-the-art MILP solvers, our method improves the quality of the solution by over 50% and reduces the computation time by 17.47%.

## 1 INTRODUCTION

Mixed-integer linear programming (MILP) has been widely used to model combinatorial optimization problems, such as production scheduling (Pochet & Wolsey, 2006), supply chain (Chao et al., 2024), energy management (Morales-España et al., 2013), and chip design (Wang et al., 2024). Existing commercial solvers (e.g., Gurobi and SCIP) are based on exact solutions, yet they are computationally expensive and fail to meet the real-time requirements of industrial applications (Morales-España et al., 2013; Li et al., 2021). Machine learning (ML), learning the relationship between the model structure and the value of the MILP solution, has become the most promising approach to solving large-scale MILPs (Bengio et al., 2021; Zhang et al., 2023; Hentenryck & Dalmeijer, 2024). Model reduction, which aims to learn a simpler model of the original MILP, provides a direct way to speed up MILP solving (Liu et al., 2024). For example, Ding et al. (2020a) and Chen et al. (2023) construct graph neural networks (GNNs) to predict a solution and fix a subset of variables to reduce the dimension of the problem (Ding et al., 2020a; Liu et al., 2025). The prediction results of partial variables can be used for smaller MIPs to build high-quality joint solutions (Nair et al., 2020; Yoon et al., 2023), or to improve local branches in the branch and bound tree search (Ding et al., 2020a; Li et al., 2018), or to identify feasible high-quality solutions efficiently (Han et al., 2023). However, most approaches to MILP reduction are based on variable reduction, overlooking constraint reduction.

Based on dual theory, constraints and variables are inherently coupled, and it is suggested that predicting and fixing constraints can also reduce search space. This paper proposes a novel constraint-based model reduction approach that transforms a subset of inequality constraints into equality constraints. As shown in Figure 1, only 5% of high-quality constraint reduction can significantly improve MILP solving performance. However, the naive reduction of inequality constraints may result in suboptimal or infeasible solutions. Therefore, the **first challenge** of constraint reduction is to identify which inequality constraints are critical so that reducing them can speed up the MILP solution. Consequently, the second challenge is how to efficiently predict these critical constraints. Existing GNN-based MILP representations focus on variable representation rather than constraint and lack the ability to predict critical constraints.

We first use the concept *tight constraint* from Operations Research (Boyd & Vandenberghe, 2004), to determine which constraints should be reduced. Given an MILP instance, tight constraints are inequality constraints that become equalities at the optimal solution. The set of tight constraints provides sufficient information necessary to achieve the optimal solution and thus is safe to reduce. However, predicting a high-dimensional vector of tight constraints is often challenging due to the complex combinatorial nature of MILP, insufficient training data, and limited model capacity. In this paper, instead of reducing all the tight constraints, we only reduce a subset of *critical* tight constraints, resulting in a significant acceleration.

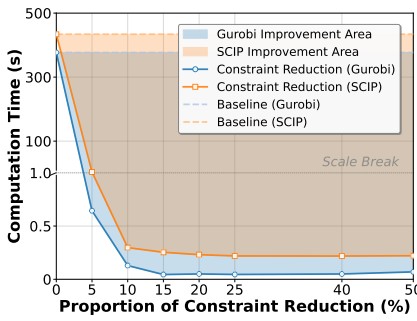

Figure 1: The results of constraint reduction on computation time on the CA_easy task, where the highest-quality constraints are selected.

In particular, we select these tight constraints that can provide the highest information gain and effectively reduce the uncertainty of the problem. Theoretical results show that these critical tight constraints can greatly reduce the feasible region of solutions, thereby accelerating MILP solving. Due to the strong correlation between critical tight constraints and constraint categories, it is necessary to incorporate the category information into the representation. To this end, we present a multi-modal representation that combines the abstract model with an instance-level model. The abstract model can capture MILP's high-level category information, so that the model can capture the structure similarity between constraints in the same category. Thus, this bi-level encoding provides a more expressive and generalizable foundation for accurately identifying critical tight constraints.

The contributions of this paper are summarized as follows. **First**, we propose a novel constraint-based model reduction framework, which predicts critical tight constraints and reduce the size of the solution space. **Second**, we introduce a novel multi-modal representation that incorporates the abstract models into its instance model, which can accurately predict critical tight constraints. **Finally**, we conduct extensive experiments on large-scale MILP domains to validate the proposed constraint-based model reduction method. Results show that our method outperforms the state-of-the-art benchmarks by over 50% in solution quality (e.g., primal gap) and improves the computation time with an average 17.47%.

## 2 RELATED WORKS

**Learning-based Model Reduction.** Model reduction is a direct approach to reducing MILP size, thereby speeding up MILP solving. Extensive research has focused on variable reduction, which predicts and fix a subset of variables to reduce the search space of solutions (Khalil et al., 2022; Paulus & Krause, 2023; Zeng et al., 2024; Chen et al., 2024; Geng et al., 2025). Fixing false-predicted variables can sometimes lead to the infeasiblity of the original MILP problem (Ding et al., 2020b; Nair et al., 2020; Chen et al., 2023). Instead of directly using the predicted solutions, threshold-based learning Yoon et al. (2023), local branching (Han et al., 2023; Huang et al., 2024a; Liu et al., 2025), and large neighborhood search (Sonnerat et al., 2021; Ye et al., 2023; 2024) are proposed to guide the search for feasible solutions. In addition to variable reduction, a novel constraint reduction framework is proposed. Bertsimas & Stellato (2022) and Li et al. (2024) have attempted to learn all tight constraints, from which the optimal solution can be obtained efficiently. However, the total number of tight constraints is very large, as it grows exponentially with system size (Misra et al.,

2022). Predicting a high-dimensional vector of tight constraints is challenging. In contrast, this paper carefully selects a subset of critical tight constraints based on their contribution to reducing the local feasible space. In the specific domain of smart grids, Porras et al. (2022); Park et al. (2023); Yang et al. (2020); Velloso & Van Hentenryck (2021) identify and remove redundant constraints (e.g., line-flow constraints) to speed up the solution of the problem. In contrast, by defining the contribution of each type of constraints to the set of tight constraints, we propose a general constraint reduction approach.

**MILP Representation.** Effective feature extraction is crucial for learning-based approaches of MILP problems. Bipartite graph is a straightforward way to capturing the structural relationships between variables and constraints (Khalil et al., 2017; Selsam et al., 2018; Gasse et al., 2019). Chen et al. (2022) theoretically shows that bipartite graphs can fully represent linear programming problems, and with random features (Chen et al., 2023), GNNs can represent MILPs sufficiently. To capture connections among the variables, constraints, and the objective, Ding et al. (2020b) proposes a tripartite graph representation and embedding method. Moreover, EGAT-based methods (Ye et al., 2024; Deng et al.) leverage edge features, offering more expressive representations. However, existing approaches focus on the representation of the MILP instance-level model, rather than the high-level abstract model that can provide uniform structural information among different instances.

## 3 THE PRELIMINARY AND OBJECTIVE

**Mixed Integer Linear Program (MILP).** The MILP can be formalized as follows:

$$\min \mathbf{c}^\top \mathbf{x} \quad \text{s.t.} \ \mathbf{A}\mathbf{x} \leq \mathbf{b} \quad \text{and} \quad \mathbf{x} \in \mathbb{Z}^q \times \mathbb{R}^{n-q}, \tag{1}$$

where $\mathbf{x} = (x_1, \ldots, x_n)^\top$ is an $n$-dimensional decision variable consisting of $q$ integer variables and $n - q$ continuous variables, $\mathbf{c} \in \mathbb{R}^n$ denotes the vector of objective coefficients, $\mathbf{A} \in \mathbb{R}^{m \times n}$ denotes the constraint coefficient matrix, and $\mathbf{b} \in \mathbb{R}^m$ is the right-hand side of the constraints. This paper focuses on binary integer variables with $\mathbf{x} \in \{0, 1\}^q \times \mathbb{R}^{n-q}$, which can be extended to general MILPs by encoding integers in binary (Nair et al., 2020). A drawback of this binary encoding is the potential increase in the problem's scale and a weaker resulting LP relaxation. For convenience, let $\mathcal{I} = \langle \mathbf{c}, \mathbf{A}, \mathbf{b} \rangle$ denote an MILP instance.

**Tight Constraints and Fixing.** Given an MILP instance $\mathcal{I}$ defined in Eq. (1), let $x^*$ denote an optimal solution. A constraint of the form $a_i^\top x \leq b_i$ is called *tight* at $x^*$ if $a_i^\top x^* = b_i$; all other constraints are redundant and can be discarded (Boyd & Vandenberghe, 2004). In our reduction framework, *fixing* a constraint means turning a tight inequality into an equality, i.e., replacing $a_i^\top x \leq b_i$ by $a_i^\top x = b_i$ in the reduced model. What's more, for a prototypical constraint category $C_i$, we say that this category is *fixed* when all its tight inequalities are transformed into equalities.

**Model Reduction.** Model reduction consists of variable and constraint reductions. Variable reduction, which is used to predict a subset of integer variables $\mathbf{x}_I \in \{0, 1\}^q$, has been extensively studied (Han et al., 2023). The constraint reduction is to identify a subset of tight constraints and transform them into equality constraints, which can also reduce the search space of feasible solutions.

**The Objective.** Given a MILP problem, although different MILP instances are not the same, only the key parameters (e.g., $\mathbf{c},\mathbf{A},\mathbf{b}$) vary slightly, and the structure remains unchanged. We can exploit the repetitive structure of the MILP instances and solutions and learn to solve unseen MILP instances. Therefore, our objective in this paper is to learn the mapping from the parameter $\langle \mathbf{c}, \mathbf{A}, \mathbf{b} \rangle$ to the reduced model as an intermediate step for MILP solving.

## 4 METHODOLOGY

The proposed model reduction framework comprises two phases: 1) *Multi-Modal Representation* to provide a more expressive foundation for predicting critical tight constraint (i.e., Section 4.1); 2) *Constraint-based Model Reduction* for labeling **C**ritical **T**ight **C**onstraints (CTCs) (i.e., Section 4.2). The overview of our framework is shown in Figure 2.

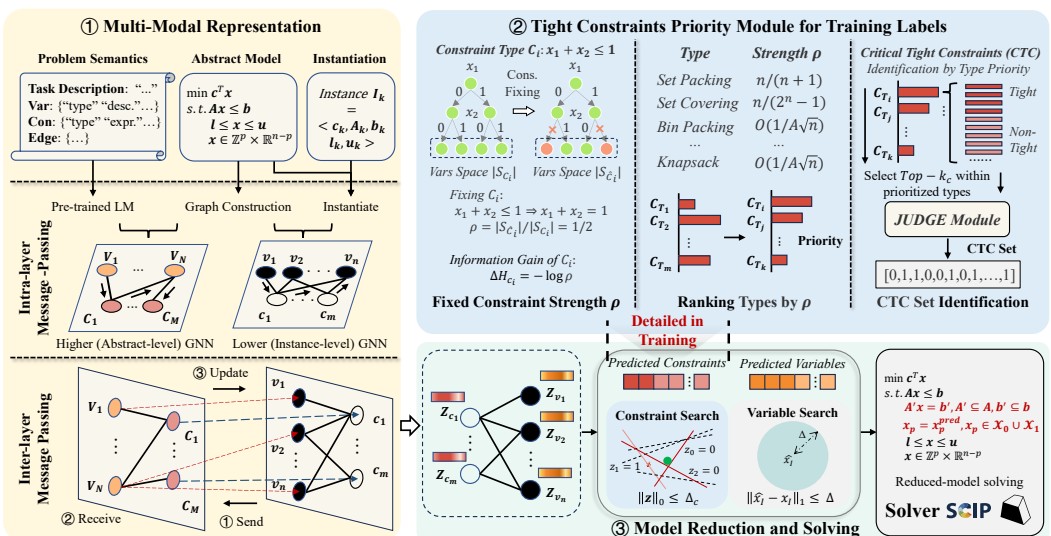

Figure 2: An overview of our framework, which comprises three components: 1) Multi-modal representation, 2) Identification of Critical Tight Constraints, and 3) Model reduction. In 1), the textual semantics are first embedded as initial features for the abstract model, after which both the instance and abstract model are transformed into bipartite graph. In 2), a subset of tight-constraints is selected and labeled as critical constraints. In 3), multi-modal representation and critical constraints are fed into the learning architecture in Section 4.1 to predict the reduced variables and constraints.

## 4.1 MULTI-MODAL REPRESENTATION

As shown in Gasse et al. (2019), a MILP instance can be represented as a weighted bipartite graph $\mathcal{G} = (\mathcal{C} \cup \mathcal{V}, \mathcal{E})$, called instance model graph. This graph has constraint and variable nodes ($\mathcal{C}$ and $\mathcal{V}$), with edges between them when a variable has a non-zero coefficient in a constraint, and edge weights equal to these coefficients. Please see Appendix F for more information on the graph features.

For a class of MILP problems $\mathcal{M}$, there exists a unified abstract model, represented as an unweighted bipartite graph $\mathcal{G}_m = (\mathcal{C}_m \cup \mathcal{V}_m, \mathcal{E}m)$ (the abstract model graph), as depicted in Figure 2. In this graph, nodes represent variable and constraint types, with an edge indicating that a variable type appears in a constraint type. The abstract model's textual descriptions, denoted as $T_m$, are converted into text embeddings using a pre-trained language model $\text{PLM}(\cdot)$, detailed in Appendix C.4. These embeddings serve as the initial features $\{\hat{h}_{V_j}^{(0)}\} = \text{PLM}(T_m)$ for the abstract model graph, where $V_j$ denotes a category node corresponding to a class of variables or constraints.

Prior work using Graph Neural Networks (GNNs) has focused on the coefficient matrix and structural properties of instance models $\mathcal{G}$, overlooking high-level abstract model information. The identification of critical tight constraints is highly correlated with constraint categories, making category-level information essential for effective representation. To this end, we regard the abstract model as a modality of MILP, introduce it to capture high-level category information.

Specifically, our model comprises two key components: (1) Intra-Layer Message Passing (MP) and (2) Inter-Layer Message Passing. Intra-layer MP refers to independent message propagation within the abstract model graph (high level) and the instance model graph (low level). In contrast, inter-layer MP facilitates information exchange between the high-level and low-level graphs, enabling effective feature fusion across different levels of representation.

**Intra-layer MP.** In the intra-layer message passing stage, the high-level GNN operates on the abstract model graph, incorporating text embeddings, while the low-level GNN follows the GNN framework proposed by Gasse et al. (2019). Let $N(v_i) \equiv v_i \in \mathcal{V} : \mathcal{A}_{i,j} \neq 0$. For convenience, we focus on a specific category node class $V_j$ and its corresponding instance nodes $v_i$, where $v_i \in V_j$ denotes that instance node $v_i$ belongs to category node $V_j$. Hence, we have

$$\hat{h}_{V_j}^{(k)} \equiv \hat{f}_2^{(k)}\Big(\big\{\hat{h}_{V_j}^{(k-1)}, \hat{f}_1^{(k)}\big(\big\{\hat{h}_U^{(k-1)} : U \in N(V_j)\big\}\big)\big\}\Big), \tag{2}$$

$$h_{v_i}^{(k)} \equiv f_2^{(k)}\Big(\big\{ h_{v_i}^{(k-1)}, f_1^{(k)}\big(\{ h_u^{(k-1)} : u \in N(v_i) \}\big)\big\}\Big), \tag{3}$$

where function $f_1^{\{k\}}$ is the neighbor aggregation function, while $f_2^{\{k\}}$ focuses on integrating neighbor information with the existing hidden features in the $k^{th}$ layer. $\hat{h}_{V_j}^{(k)}$ denotes the higher GNN's features while $h_{v_i}^{(k)}$ denotes lower GNN's features.

**Inter-layer MP.** In the inter-layer message passing stage, this process consists of three phases: *Sending*, *Receiving* and *Updating*.

In the *sending* phase, for category node $V_j$ in the abstract model, we extract the corresponding set of nodes $\{v_i\}$ from the instance model. Cross-Attention (Bahdanau et al., 2014) is then applied to achieve cross-modal fusion, resulting in instance-category features $\tilde{h}_{V_j}^{(k)}$ enhanced with instance model node features, which can be calculated by:

$$\tilde{h}_{V_j}^{(k)} = \text{CrossAttention}(\hat{h}_{V_j}^{(k)}, \text{MLP}(\{h_{v_i}^{(k)}\}), \text{MLP}(\{h_{v_i}^{(k)}\})), \quad v_i \in V_j. \tag{4}$$

In the *receiving* phase, category nodes $V_j$ in the abstract model receive the instance-category features $\tilde{h}_{V_j}^{(k)}$ obtained from Eq. (4). After concatenation, these features pass through an MLP layer to derive new and more enriched category features $\hat{h}_{V_j}^{(k)}$,

$$\hat{h}_{V_j}^{(k)} = \text{MLP}(\hat{h}_{V_j}^{(k)} \parallel \tilde{h}_{V_j}^{(k)}). \tag{5}$$

In the *updating* phase, we represent the category features in the instance model with the average instance node features, $\bar{h}_{V_j}^{(k)} = \text{mean}(\{h_{v_i}^{(k)}\}), v_i \in V_j$. To balance the category features from both the instance and abstract models, we introduce a gating mechanism, which gets a dynamic weight for category features. Here, $\text{Gate}(\cdot) = \text{MLP}(\cdot)$. Finally, we fuse the resulting category features with the node features in the instance model $\{h_{v_i}^{(k)}\}$ to obtain a more comprehensive representation. The aforementioned computation can be formulated as follows:

$$\alpha = \text{Gate}(\bar{h}_{V_j}^{(k)} \parallel \hat{h}_{V_j}^{(k)}), \quad \alpha \in [0,1], \tag{6}$$

$$h_{v_i}^{(k)} = (\alpha \bar{h}_{V_j}^{(k)} + (1-\alpha)\hat{h}_{V_j}^{(k)}) \odot h_{v_i}^{(k)}, \quad v_i \in V_j, \tag{7}$$

where $\odot$ denotes Hadamard product and $\parallel$ denotes concatenation. We conducted a complexity analysis of the model in the appendix. For details, please refer to the appendix E.

## 4.2 MODEL REDUCTION

**Observation.** We were curious about the absence of existing work on directly predicting and fixing constraints to reduce model. Therefore, we conducted experiments to evaluate the effectiveness of this approach. We first introduce tight constraints, fixing them without affecting the problem's optimality. Then we randomly fix tight constraints with ground truth values and it leads to highly variable solving times across runs. Some cases solve much faster than direct solving, while others are slower. The results for toy datasets in Table 1 showed in worst cases, it even severely deteriorated the solving

Table 1: We conducted motivation experiments via fixing different tight constraints on two datasets. Please refer Appendix H.1 to see details of this experiment.

| Time (s) | CA_easy | WA |
|---|---|---|
| Original | 378.226 | >3600 |
| Best Fixing | 1.851 | 50.726 |
| Worst Fixing | 465.74 | >3600 |

time, which suggests that fixing different tight constraints has varying impacts on solving acceleration. Based on the above experiments and related empirical evidence on ML-based prediction, we attribute the failure of directly predicting and fixing tight constraints to the following two reasons:

(1) **Critical Tight Constraints (CTCs)**, which are capable of accelerating the solving process, exhibit varying impacts due to differences in the structural forms and functional roles of constraints.

(2) **Selective Learning Bias**. Neural networks often assign high confidence to simple constraints, which are typically non-critical, resulting in poor fixings and reduced solver efficiency.

Therefore, to leverage constraints for accelerating the solving process, we propose a method to identify and fix critical tight constraints addressing the two aforementioned challenges.

**Identification of CTC.**   In this paragraph, we will introduce a **T**ight **C**onstraints **P**riority (TCP)—an information theory-inspired heuristic module for identifying critical tight constraints and setting training labels. First, we select 10 of the 17 basic constraint types from MIPLIB (Gleixner et al., 2021) as prototypical constraints (covering nearly all MILP constraint types), excluding equation constraints, logical constraints, and other less common ones. These 10 types are shown in Table 2.

Table 2: The 10 selected prototypical constraints and their estimated strengths. $\sim$ indicates that the strength is hard to uniformly quantify and represent. See Appendix B.2 for the derivation.

| Type | Expression | Strength ($\rho$) |
|---|---|---|
| Singleton | a single variable | $\sim$ |
| Precedence | $ax - ay \leq b$, where $x$ and $y$ have the same type | $\sim$ |
| Variable Bound | $ax + by \leq c, x \in \{0,1\}$ | $\sim$ |
| Set Packing | $\sum x_i \leq 1, x_i \in \{0,1\}, \forall i$ | $n/(n+1)$ |
| Set Covering | $\sum x_i \geq 1, x_i \in \{0,1\}, \forall i$ | $n/(2^n - 1)$ |
| Invariant Knapsack | $\sum x_i \leq b, x_i \in \{0,1\}, \forall i, b \in \mathbb{N}_{\geq 2}$ | $O(1/\sqrt{n})$ |
| Bin Packing | $\sum a_i x_i + ax \leq a, x, x_i \in \{0,1\}, \forall i, a \in \mathbb{N}_{\geq 2}$ | $O(1/A\sqrt{n})$ |
| Knapsack | $\sum a_i x_i \leq b, x_i \in \{0,1\}, \forall i, b \in \mathbb{N}_{\geq 2}$ | $O(1/A\sqrt{n})$ |
| Mixed Binary | $\sum a_i x_i + \sum p_j s_j \leq b, x_i \in \{0,1\}, \forall i, s_j$ cont. $\forall j$ | $\sim$ |
| General Linear | no special structure | $\sim$ |

**Definition 1** (Local Feasible Space $S_{C_i}$). *For a prototypical constraint $C_i$ (a type of constraint) acting on a set of $k$ variables $X_{C_i} := \{x_1, \ldots, x_k\}$, we define the Variable Domain Space as $\mathcal{D}_i := \prod_{j=1}^{k} dom(x_j), x_j \in X_{C_i}$, where $dom(\cdot)$ denotes the domain of a variable. Thus, we have:*

$$S_{C_i} := \{\mathbf{x} \in \mathcal{D}_i \mid C_i(\mathbf{x}) \text{ is satisfied}\}. \tag{8}$$

*Here, "$C_i(\mathbf{x})$ is satisfied" means that, for a given assignment $\mathbf{x} \in \mathcal{D}_i$ to the variables $X_{C_i}$, all constraint conditions belonging to the type $C_i$ are simultaneously fulfilled under $\mathbf{x}$, i.e., none of the constraints in this category is violated.*

**Definition 2** (Fixed Constraint Strength $\rho$). *We use $\hat{C}_i$ to denote the set of constraints after fixing the tight constraints in the constraint category $C_i$. We define the fixed constraint strength as:*

$$\rho(C_i) := \frac{|S_{\hat{C}_i}|}{|S_{C_i}|}, \quad \rho \in [0,1]. \tag{9}$$

$|\cdot|$ denotes the size of a set. Hereinafter, $\rho(C_i)$ is abbreviated as $\rho$.

We define the fixed constraint strength $\rho$ in Eq. (9). We note that this strength is primarily related to the structural form of the constraint. Therefore, we estimate its strength by analyzing the prototypical form of a class of constraints. The inherent coupling of constraints in MILPs renders the analysis of their individual impacts intractable. We therefore introduce a local decoupling simplification. This positions our theoretical framework as a principled heuristic guided by fixed constraint strength $\rho$, whose validity is ultimately corroborated by robust empirical results.

**Proposition 3** (Entropy-Driven Acceleration). *Let $P(X_{C_i}|I)$ represent the probability distribution of the values taken by the variables acted upon by the prototypical constraint $C_i$. Thus, we have assumption:*

$$P(X_{C_i}|I) = P(X_{C_i}|C_i, \mathcal{U}) \approx P(X_{C_i}|C_i), \tag{10}$$

*where $\mathcal{U}$ denotes the influence noise of other constraint types in the instance $I$ on the $X_{C_i}$. Therefore, suppose (10) holds [1], We have the local information gain $\Delta H_{C_i}$ of fixing $C_i$ as below:*

$$\Delta H_{C_i} = -\log \rho. \tag{11}$$

*We interpret the feasible region of a MILP as a space of uncertainty, and fixing constraints can be viewed as a process of entropy reduction. We have that a lower $\rho$ value implies a larger reduction in uncertainty, which in turn effectively shrinks the search space during solving and accelerates the process—this partly demonstrates the validity of our heuristic method based on $\rho$.*

---

[1] For this Assumption, there is to show that the method admits favorable theoretical properties when the MILP exhibits certain structural characteristics. Without these characteristics, our proposed method alsp remains effective, please refer to Appendix H.7 for more discussion.

Please refer to Appendix B.1 for the proof. Therefore, we identify critical tight constraints based on the pre-estimated strength $\rho$ of prototypical constraints and the structural form of each constraint. Specifically, we prioritize constraints from categories with a lower $\rho$, which significantly reduce the local feasible region, yielding high information gain $\Delta H_{C_i} = -\log \rho$ for the problem. Next, we treat the tight constraints in these categories as critical tight constraints and conduct learning on them. To maximize sample balance, we perform coarse-grained selection of critical tight constraints in TCP module. See Algorithm 1 for TCP module details.

**Model Reduction.** Given a MILP instance $\mathcal{I}$ belonging to a problem class $\mathcal{M}$, let $\theta$ denote the parameters of our proposed network. The model generates predictions $p_\theta(\mathbf{c}, \mathbf{x} \mid \mathcal{I}, \mathcal{M})$ to identify critical components. To handle predicted tight constraints, we introduce a hyperparameter $k_c$ to select the top-ranked constraints and a trust region parameter $\Delta_c$ to allow for flexible adjustments. This trust region mechanism is crucial for mitigating potential infeasibility or suboptimality caused by prediction errors. For integer variable prediction and search, we follow the methodology proposed by Han et al. (2023), which is detailed in Appendix C.5.

Specifically, we reduce the original MILP to the following formulation, where $I_c$ represents the set of indices for the $k_c$ selected constraints, $M$ is a sufficiently large constant (Big-M) and $z_i \in \{0, 1\}$ are auxiliary indicator variables. In practice, this logic can also be efficiently implemented via the solver's native indicator constraints:

$$
\begin{aligned}
\min \quad & \mathbf{c}^\top \mathbf{x} \\
\text{s.t.} \quad & \mathbf{A}\mathbf{x} \leq \mathbf{b}, \\
& a_i^\top \mathbf{x} \geq b_i - M z_i, \quad \forall i \in I_c, \\
& \sum_{i \in I_c} z_i \leq \Delta_c, \\
& \mathbf{x} \in \mathcal{D} \cap \mathcal{B}(\mathcal{X}_0, \mathcal{X}_1, \Delta).
\end{aligned}
\tag{12}
$$

The rationale behind this reduction is that fixing (or penalizing the violation of) tight constraints preserves the original optimal solution $x^*$, as $x^*$ inherently satisfies these constraints. By restricting the search space to a region that remains likely to contain $x^*$, we maintain the feasibility and optimality of the original problem while significantly reducing the computational burden. Even in the presence of prediction noise, the trust region $\Delta_c$ ensures that the reduced feasible region remains a non-empty subset of the original space that is highly likely to retain the optimal guarantee.

**Focal Loss.** To mitigate the second issue—selective learning bias from the imbalance between easy and hard samples—we use Focal Loss (Lin et al., 2017) instead of Cross-Entropy Loss.

$$
\mathcal{L}_{\text{Focal}} = -\alpha(1 - \hat{y}_i)^\gamma y_i \log \hat{y}_i + (1 - \alpha)\hat{y}_i^\gamma (1 - y_i) \log(1 - \hat{y}_i),
\tag{13}
$$

where $y_i$ is the true label, $\hat{y}_i$ is the predicted probability, $\alpha$ is a weighting factor for class balance, and $\gamma$ is a parameter that adjusts the weighting of hard and easy samples. The following is our loss:

$$
\mathcal{L}(\theta) = \lambda \mathcal{L}_{\text{Focal}}^{\text{sol}} + (1 - \lambda) \mathcal{L}_{\text{Focal}}^{\text{con}},
\tag{14}
$$

where $\mathcal{L}_{\text{Focal}}^{\text{sol}}$ denotes the Focal Loss for predicting variable assignments and $\mathcal{L}_{\text{Focal}}^{\text{con}}$ denotes the Focal Loss for identifying critical tight constraints.

## 5 EXPERIMENTS

In this section, to demonstrate the advantages of the proposed framework, we conducted comprehensive computational studies on (i) solving performance, (ii) generalization ability, and (iii) compatibility with real-world scenarios. In addition, to verify the effectiveness of each component within the proposed framework, we conducted extensive ablation studies. Additional numerical results and implementation details can be found in the Appendix H.

### 5.1 EXPERIMENTAL SETUP

**Benchmarks.** Our evaluation is carried out on four widely adopted benchmark datasets: the maximum independent set (MIS), minimum vertex cover (MVC) (Albert & Barabási, 2002), combinatorial

auction (CA) (Leyton-Brown et al., 2000), and workload appointment (WA) (Gasse et al., 2022) problem. Specifically, the first two datasets (MIS and MVC) represent graph optimization problems, while the latter two (CA and WA) correspond to non-graph optimization scenarios. These datasets are widely adopted in (Huang et al., 2024a; Han et al., 2023; Liu et al., 2025; Gasse et al., 2019) for benchmarking and empirical evaluation. A more detailed description of the datasets and their generation processes can be found in the Appendix C.

**Baselines.** In our experimental setup, we consider the following baselines for comparison: (i) traditional solvers, including SCIP 8.0.1 (Achterberg, 2009) and Gurobi 9.5.2 (Gurobi Optimization, 2022), (ii) the Predict-and-Search (PS) method (Han et al., 2023), as described in Appendix C.5, and (iii) the Contrastive Predict-and-Search (ConPaS) method (Huang et al., 2024a), which is employed as a strong baseline to demonstrate the performance of our approach. It differs from PS in that it uses contrastive learning and leverages it to improve the quality of predicted solutions. For ConPaS, we set the ratio of positive to negative samples at ten, using low-quality solutions as negative samples. By including negative samples in the training process, ConPaS can better distinguish between high-quality and low-quality solutions.

**Metrics.** We use the following metrics in the experiment: (i) *Relative primal gap* (Berthold, 2006), which shows the difference between the primal objective value $z$ and the best known objective value (BKS) $z^*$. It can be defined as $\frac{|z-z^*|}{max(z,z^*,\epsilon)}$, where $\epsilon$ is a small positive value to avoid the numerical problem. We follow the setting in Han et al. (2023), running a Gurobi single-thread for $3,600$ seconds to get $z^*$. (ii) *Absolute primal gap* (called $\text{gap}_{\text{abs}}$) is the difference between the best objective found by the solvers and BKS, defined as $\text{gap}_{\text{abs}} = |z - z^*|$. (iii) *The primal-dual integral* (called PDI) (Berthold, 2013) is defined as the integral of the primal-dual gap, with lower values indicating better convergence. Let $z_D$ be the dual objective bound, and the primal-dual gap is defined as $\text{gap} = \frac{|z-z_D|}{|z|}$. Our method is a model reducer: it takes an original MILP as input and outputs a "simpler" reduced one. A good reduced MILP should help an off-the-shelf solver perform better, with better quality measured by faster convergence speed (via PDI) and higher solution quality (via gap).

**Configurations.** We conduct experiments on AMD Ryzen 9 9900X 12-Core Processor. All evalutions are performed under the same configurations. We set MIPFocus = 1 for Gurobi and enabled the SCIP_PARAMSETTING.AGGRESSIVE mode for SCIP, both of which run in a single-thread. Following Han et al. (2023), we use 240 trairaging, 60 validation, 100 testing instances. And for testing, we set the run time cut off to 800 seconds to solve the reduced MILP of each test instance. For more details, please refer to the Appendix G.

## 5.2 EVALUATION RESULTS

To assess the efficacy of the proposed approach, we contrast the solving performance of our framework with that of the baselines, within a time constraint of 800 seconds. Table 3 provides a comprehensive overview of the experimental results from two dimensions. $\text{gap}_{\text{abs}}$ measures the solution quality, and PDI quantifies the overall convergence behavior of the optimization process. Overall, ConPaS slightly outperforms the PS in terms of solution quality, while the two methods exhibit comparable convergence speeds. Our proposed method surpasses both approaches in both solution quality and convergence speed. Across all methods utilizing Gurobi and SCIP, our proposed method achieves an average improvement of 51.06% in $\text{gap}_{\text{abs}}$ and 17.47% in PDI over the best-performing baseline.

In addition, Figure 3 presents the evolution of the primal gap over time using SCIP, where a faster decline in the curves reflects better solving performance. As shown in Figure 3, our method demonstrates a more rapid decrease in the primal gap compared to other methods, highlighting its superior efficiency in solving MILP problems within the 800s time limit. Moreover, the primal gap of our method consistently outperforms those of the ConPaS and PS methods. We have also reported the experiments of all baselines based on Gurobi and hyperpapameters analysis in the Appendix H.

**Generalization.** To assess the generalization capability of our model, we further conduct evaluations on larger versions of the CA and MVC datasets. Detailed dataset scales and statistics are reported in the Appendix C.1. The results are shown in Table 4. It can be observed that all machine learning-

Table 3: Comparison of Different Methods on Four Datasets (800s time limit). The last column shows our method's improvement over the traditional solver (↓ means lower is better). The best are in **bold**.

| Method | CA | | MIS | | MVC | | WA | |
|---|---|---|---|---|---|---|---|---|
| | $\text{gap}_{abs}$ ↓ | PDI ↓ | $\text{gap}_{abs}$ ↓ | PDI ↓ | $\text{gap}_{abs}$ ↓ | PDI ↓ | $\text{gap}_{abs}$ ↓ | PDI ↓ |
| Gurobi | 477.51 | 76.29 | 2.70 | 33.09 | 8.38 | 68.02 | 0.20 | 4.76 |
| PS+Gurobi | 210.19 | 73.78 | 0.04 | 0.69 | 0.28 | 7.39 | 0.07 | 4.97 |
| ConPaS+Gurobi | 197.36 | 73.75 | 0.02 | 0.71 | 0.10 | 7.64 | 0.10 | 4.97 |
| **Ours+Gurobi** | **104.72** | **73.02** | **0.00** | **0.47** | **0.00** | **6.26** | **0.06** | **4.40** |
| **Improvement (%)** | 77.06% | 4.22% | 100.00% | 98.58% | 100.00% | 90.80% | 70.00% | 7.56% |
| SCIP | 4005.99 | 190.18 | 4.16 | 103.91 | 12.42 | 140.89 | 2.60 | 37.06 |
| PS+SCIP | 3575.18 | 179.48 | 0.66 | 6.49 | 2.98 | 21.03 | 1.27 | 34.85 |
| ConPaS+SCIP | 3545.92 | 146.88 | 0.43 | 5.52 | 2.56 | 24.05 | 1.20 | 29.91 |
| **Ours+SCIP** | **3401.63** | **109.41** | **0.34** | **4.65** | **1.48** | **18.34** | **0.83** | **21.52** |
| **Improvement (%)** | 15.08% | 42.47% | 91.83% | 95.52% | 88.08% | 86.98% | 68.08% | 41.93% |

based methods exhibit good generalization capabilities. Notably, our method continues to achieve the best solution quality and convergence speed even in larger-scale generalization experiments.

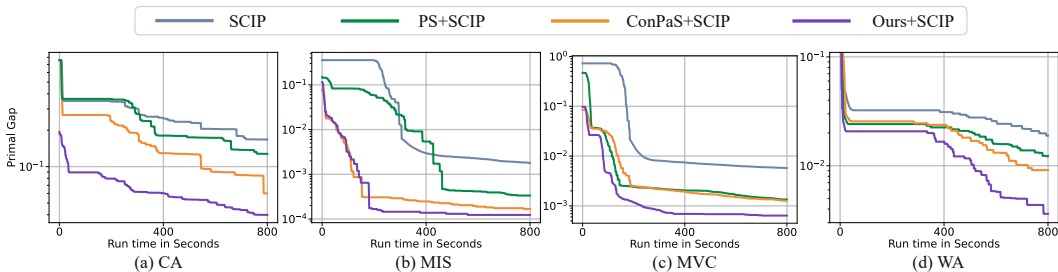

Figure 3: The relative primal gap (the lower the better) as a function of runtime, averaged over 100 test instances, within 800s time limits.

**Real-world Scenarios.** One limitation of our approach is its reliance on explicit mathematical models and problem descriptions, preventing direct evaluation on standard benchmarks like MIPLIB (which often lacks such details). Instead, we use the real-world Middle-Mile Consolidation Network (MMCN) (Huang et al., 2024b) to assess its practical performance. As an alternative, we select the Middle-Mile Consolidation Network (MMCN) in Huang et al. (2024b), a set of real-world problem instances to assess the practical performance of our approach. Please refer to the appendix C.2 for the detailed information of this dataset. The solving results between baselines and our method on this dataset are presented in Table 5.

Table 4: Generalization results on larger-scale CA and MVC with SCIP within 800s across 100 test instances. ↓ indicates that lower is better. The best are in **bold**.

| Method | CA | | MVC | |
|---|---|---|---|---|
| | $\text{gap}_{abs}$ ↓ | PDI ↓ | $\text{gap}_{abs}$ ↓ | PDI ↓ |
| SCIP | 7009.58 | 198.78 | 22.07 | 228.21 |
| PS | 6541.87 | 169.86 | 4.07 | 39.63 |
| ConPaS | 6287.22 | 156.95 | 2.20 | 35.39 |
| Ours | **5955.86** | **131.44** | **0.55** | **24.54** |

Table 5: Results on the real-world dataset MMCN (Huang et al., 2024b), using SCIP within 800s over 100 test instances.

| Method | MMCN | |
|---|---|---|
| | $\text{gap}_{abs}$ ↓ | PDI ↓ |
| SCIP | 13819.90 | 427.63 |
| PS | 8084.36 | 329.60 |
| ConPaS | 6064.85 | 330.05 |
| Ours | **3547.08** | **246.89** |

**Ablation Study.** To better understand each component's contribution, we conduct an ablation study on the CA dataset using SCIP (800s time limit, 100 test instances). First, we use GNN to predict and fix variables/constraints ("w/o Multi-Modal") to highlight constraint fixing effectiveness. Second, we use only multimodal representation ("w/o Constraints") to show its contribution to prediction. Figure 4 shows both variants outperform PS but are slightly inferior to our complete approach. Before the time limit, our multimodal representation consistently shows significant benefits, despite negligible gain at the end. For example, at 400s marked, our full method improves the primal gap by 12% over

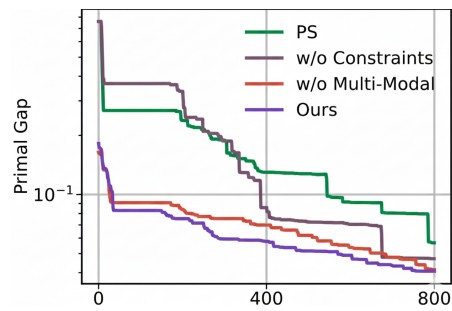

Figure 4: Ablation study results: the relative primal gap as a function of runtime.

"w/o Multi-Modal". For more analysis on the "multi-modal representation", refer to Appendix H.5.

## 6 CONCLUSION

This paper proposes a constraint-based model reduction method by reducing both variables and constraints. There are two challenges for constraint reduction: 1) which inequality constraints are critical, and 2) how to predict these critical constraints efficiently. This paper first labels a subset of tight constraints according to their contribution to reducing the search space. Furthermore, a multi-modal representation is proposed to encode both instance-level and abstract-level MILP, which can fully capture the category information of constraints. Simulations on real-world problems show that the proposed method has a significant improvement in computation time and solution quality.

## 7 ACKNOWLEDGEMENTS

This research was supported by National Key Research and Development Program of China (NO. 2024YFB4303805), the Key Research and Development Projects in Jiangsu Province (No.BE2021001-2), the National Natural Science Foundation of China (Nos. 62476121). The work was a collaborative effort jointly conducted by School of Computer Science and Engineering, Southeast University and Huawei Noah's Ark Lab. We appreciate the close collaboration, technical discussions, and shared resources from both teams that made this joint research possible.

## 8 REPRODUCIBILITY STATEMENT

To ensure the reproducibility of our research, we provide all necessary implementation details. The key source code has been uploaded as an attachment to this submission. We further commit to releasing the entire source code on a public GitHub repository upon acceptance of the paper. The datasets used in this study are all publicly available, with detailed descriptions, sources, and preprocessing steps provided in Appendix C.1. The specific experimental setup and computational resources are described in Section 5.1. All hyperparameter settings for our models are comprehensively listed in Appendix G and D.

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

## A  THE USE OF LARGE LANGUAGE MODELS

Large Language Models (LLMs) were utilized to assist in generating the JSON text required for this work and to support writing revisions. Specifically, we leveraged an LLM, providing it with detailed descriptions of the relevant problems and corresponding mathematical models, along with specified output formats, to generate the JSON content we needed. Additionally, the LLM aided in tasks such as refining language expression, enhancing text readability, and ensuring clarity in various sections of the paper, including sentence rephrasing, grammar checking, and improving the overall textual flow.

It is important to note that the LLM was not involved in the ideation, research methodology, or experimental design. All research concepts, ideas, and analyses were developed and conducted independently by the authors. The contribution of the LLM was solely focused on aspects related to text generation and linguistic polishing, with no involvement in the scientific content or data analysis.

The authors take full responsibility for the entire content of the manuscript, including any text generated or polished by the LLM. We have ensured that the text produced with the assistance of the LLM complies with ethical guidelines and does not lead to plagiarism or scientific misconduct.

## B  THEORETICAL PROOFS

### B.1  PROOF OF PROPOSITION 3

According to the Maximum Entropy Principle, we define the distribution $P(X_{C_i}|I)$ as the uniform distribution over all variable assignments permitted by constraint $C_i$. And based on Assumption 10, we have:

$$P(X_{C_i}|I) \approx P(X_{C_i}|C_i) = \frac{1}{|S_{C_i}|},$$
$$P(X_{C_i}|I') \approx P(X_{C_i}|\hat{C}_i) = \frac{1}{|S_{\hat{C}_i}|}, \tag{15}$$

where $I'$ denotes the instance after fixing the tight constraints in the constraint category $C_i$. Thus:

$$\Delta H_{C_i} = H(P(X_{C_i}|I')) - H(P(X_{C_i}|I))$$
$$\approx H(P(X_{C_i}|\hat{C}_i)) - H(P(X_{C_i}|C_i)) \tag{16}$$
$$= -\log\left(|S_{\hat{C}_i}|\right) + \log\left(|S_{C_i}|\right) \tag{17}$$

Therefore:

$$\Delta H_{C_i} = \log\left(\frac{|S_{C_i}|}{|S_{\hat{C}_i}|}\right) = -\log \rho \tag{18}$$

### B.2  PROOF OF FIXED STRENGTH FOR 10 PROTOTYPICAL CONSTRAINTS IN TABLE 2

#### B.2.1  SET PACKING

For Set Packing $\sum x_i \leq 1, x_i \in \{0,1\}, \forall i$, we can easily have $|\mathcal{S}_{C_i}| = n+1, |\mathcal{S}_{\hat{C}_i}| = n$. Therefore,

$$\rho = \frac{|\mathcal{S}_{\hat{C}_i}|}{|\mathcal{S}_{C_i}|} = \frac{n}{n+1}.$$

#### B.2.2  SET COVERING

Consider the Set Covering constraint $\sum x_i \geq 1$ with $x_i \in \{0,1\}$ for $i = 1, \ldots, n$. The number of feasible combinations before fixing is $|\mathcal{S}_{C_i}| = 2^n - 1$, excluding the all-zero vector. After fixing the constraint to $\sum x_i = 1$, only $n$ combinations remain, corresponding to vectors with exactly one non-zero entry. Thus, the fixed constraint strength is:

$$\rho = \frac{n}{2^n - 1}.$$

### B.2.3 INVARIANT KNAPSACK

Consider the Invariant Knapsack constraint defined as $\sum_{i=1}^{n} x_i \leq b$, where $x_i \in \{0,1\}$ and $b \in \mathbb{N}_{\geq 2}$. The total number of feasible solutions is given by

$$|\mathcal{S}_{C_i}| = \sum_{k=0}^{b} \binom{n}{k},$$

which corresponds to all binary vectors of length $n$ with Hamming weight at most $b$. Among them, the tight feasible region, where the constraint is exactly satisfied, i.e., $\sum_{i=1}^{n} x_i = b$, so we have

$$|\mathcal{S}_{\hat{C}_i}| = \binom{n}{b}.$$

Therefore,

$$\rho = \frac{|\mathcal{S}_{\hat{C}_i}|}{|\mathcal{S}_{C_i}|} = \frac{\binom{n}{b}}{\sum_{k=0}^{b} \binom{n}{k}}.$$

We analyze the behavior of $\rho$ under two regimes. First, when $b = \frac{n}{2}$, the binomial coefficient $\binom{n}{b}$ achieves its maximum. Using Stirling's approximation, we obtain

$$\binom{n}{n/2} \sim \frac{2^n}{\sqrt{\pi n/2}}, \quad \sum_{k=0}^{n/2} \binom{n}{k} \sim 2^{n-1},$$

which yields

$$\rho \sim \frac{2^n/\sqrt{\pi n/2}}{2^{n-1}} = \frac{2}{\sqrt{\pi n/2}} = O\left(\frac{1}{\sqrt{n}}\right).$$

Second, when $b \sim \sqrt{n}$, the summation in the denominator has at most $b+1$ terms, each upper bounded by $\binom{n}{b}$. This implies

$$\sum_{k=0}^{b} \binom{n}{k} \leq (b+1)\binom{n}{b} \quad \Rightarrow \quad \rho \geq \frac{1}{b+1} = \Omega\left(\frac{1}{\sqrt{n}}\right).$$

Therefore, in both cases we conclude that

$$\rho = \Theta\left(\frac{1}{\sqrt{n}}\right),$$

indicating that the relative size of the tight feasible region decays with increasing $n$. Hence, the fixing strength based on tight solutions for the Invariant Knapsack constraint is asymptotically weak.

### B.2.4 KNAPSACK

Consider the classic Knapsack constraint of the form $\sum_{k=1}^{n} a_k x_k \leq b$, where $x_k \in \{0,1\}$ and $a_k > 0$. To ensure the constraint is non-trivial and informative, we assume the right-hand side $b$ is set near the mean of the total weight, i.e.,

$$b \sim \mu = \frac{1}{2} \sum_{k=1}^{n} a_k.$$

If $b$ is too small, the feasible region may collapse; if too large, the constraint becomes inactive. Thus, setting $b \sim \mu$ corresponds to the regime of maximal informativeness.

Assuming that each $x_k$ is an independent Bernoulli random variable with success probability $0.5$, the total weighted sum

$$X = \sum_{k=1}^{n} a_k x_k$$

is a sum of independent random variables. By the Central Limit Theorem, for sufficiently large $n$, the distribution of $X$ approximates a Gaussian:

$$X \sim \mathcal{N}\left(\mu = \frac{1}{2}\sum_{k=1}^{n} a_k, \ \sigma^2 = \frac{1}{4}\sum_{k=1}^{n} a_k^2\right).$$

Fixing the constraint (i.e., forcing the inequality to hold at equality or within a small margin) corresponds to conditioning $X$ to lie within a narrow band around $\mu$, for instance:

$$X \in [\mu - \delta, \ \mu + \delta],$$

where $\delta$ is a small positive constant. The probability mass within this interval under the Gaussian approximation provides an estimate of the tightness ratio $\rho$, given by:

$$\rho = \int_{\mu-\delta}^{\mu+\delta} \frac{1}{\sqrt{2\pi}\sigma} e^{-\frac{(x-\mu)^2}{2\sigma^2}} dx \leq \frac{2\delta}{\sqrt{2\pi}\sigma}.$$

This yields the upper bound:

$$\rho = O\left(\frac{1}{\sqrt{\sum_{k=1}^{n} a_k^2}}\right).$$

If the weights $a_k$ have average magnitude $A$ (i.e., $a_k \sim A$), then:

$$\sum_{k=1}^{n} a_k^2 \sim nA^2 \quad \Rightarrow \quad \rho = O\left(\frac{1}{A\sqrt{n}}\right).$$

In the most informative regime where $b \sim \mu$, the tightness ratio $\rho$ of the Knapsack constraint decays at least as fast as $O(1/\sqrt{n})$, and potentially faster depending on the weight magnitudes. Thus, similar to the Invariant Knapsack, the Knapsack constraint also becomes asymptotically weak as $n$ increases.

### B.2.5 BIN PACKING AND MIXED BINARY

Both *Bin Packing* and *Mixed Binary* constraints share similar mathematical forms with general Knapsack constraints, typically expressed as $\sum a_k x_k \leq b$ with binary or partially binary variables. Overall, *Bin Packing*'s $\rho$ values can be considered of similar magnitude to Knapsack constraints. However, for *Mixed Binary*, considering that there are some continuous variables, it is difficult to analyze them, and generally we do not consider fixing them.

### B.2.6 OTHERS

Constraints such as Singleton, Precedence, Variable Bound, and General Linear are challenging to quantify uniformly. In this work, we generally do not take these constraints into account.

## C DETAILS ON THE BENCHMARKS

In this section, we will introduce the benchmark problems. For the CA problem, there are two types of constraints that are structurally similar; however, the average number of variables in the second type is significantly smaller than in the other. This results in a smaller fixed strength $\rho$, leading to a more effective reduction in the solution space, and thus we choose to fix this type of constraint. For the MIS and MVC problems, which involve only one type of constraint, our approach also remains effective and demonstrates superior performance. In the case of the WA problem, since only the first constraints includes binary variable, we opt for this type of constraint.

### C.1 DATASET DETAILS

Table 6 shows the relevant scale information of the benchmark used in our paper. For the CA and MIS datasets, we generated them using the same method as in Gasse et al. (2019). Specifically, we used

algorithm in Leyton-Brown et al. (2000) with 2,000 items and 4,000 bids to generate CA instances. For the MVC dataset, we generated the instances according to the Barabasi-Albert random graph model Albert & Barabási (2002), with 4,000 nodes and an average degree of 5 subsequently. The WA dataset is token from the NeurIPS 2021 ML4CO competition Gasse et al. (2022), which also includes an item placement (IP) dataset that is often employed in the field. We, however, chose not to use IP as it predominantly consists of equality constraints, with a scarcity of tight constraints, thus making it unsuitable for our method.

Table 6: The average numbers of variables and constraints of the benchmarks we used in this paper.

|                      | CA   | MIS   | MVC   | WA    | MMCN    |
|----------------------|------|-------|-------|-------|---------|
| Binary Variables     | 4000 | 6000  | 4000  | 1000  | 7441.34 |
| Continuous Variables | 0    | 0     | 0     | 60000 | 529.37  |
| Constraints          | 2675 | 25549 | 19975 | 64306 | 6385.27 |

We generated CA and MVC datasets of a larger scale to verify the generalization ability of our method. Specifically, for the CA dataset, we used the same method to generate instances with 3,000 items and 6,000 bids. For the MVC dataset, we generated instances with 6,000 nodes and an average degree of 5.

For the training data collection, We use Gurobi to run for 3600 seconds, saving 50 solutions along with their corresponding tight constraint sets as labels. Notably, the time cost for collecting tight constraint labels is consistent with that of collecting solution labels alone, without introducing any additional time overhead. Additionally, during both training and testing, performing constraint prediction and reduction simultaneously incurs no extra time cost.

## C.2 REAL-WORLD DATASET

The Middle-Mile Consolidation Network (MMCN) problem is a network design challenge that focuses on developing load consolidation plans for transporting shipments from stocking points—such as vendors and fulfillment centers—to last-mile delivery destinations. The objective is to determine a minimum-cost allocation of transportation capacity across the network arcs, while ensuring that all shipment lead-time constraints are satisfied. The detailed mathematical model can be found in Greening et al. (2023). Taking into account both the computational difficulty and the model scale, we opted for MMCN Very-hard-BC2, which is available for download on the `https://sites.google.com/usc.edu/distributional-miplib/middle-mile-consolidation-network-design` (Huang et al., 2024b). We also use 240 trainging, 60 validation, 100 testing instances, keeping other settings unchanged.

## C.3 PROBLEM MILP FORMULATIONS

In the CA problem, given $n$ bids $\{(B_i, p_i) : i \in [n]\}$ for $m$ items, where $B_i$ is a subset of items and $p_i$ is the bidding price for $B_i$, $S_k$ denotes the substitutable bid group for bidder $k$, where $k \in [l]$ and we want to allocate items to bids such that the total revenue is maximized:

$$\min - \sum_{i \in [n]} p_i x_i$$
$$\text{s.t.} \sum_{i:j \in B_i} x_i \le 1, \quad \forall j \in [m],$$
$$\sum_{k \in S_i} x_k \le 1, \quad \forall i \in [l],$$
$$x_i \in \{0, 1\}, \quad \forall i \in [n].$$

In the MVC problem, given an undirected graph $G = (V, E)$ with a weight $w_v$ associated with each node $v \in V$, we want to select a subset of nodes $V' \subseteq V$ with the minimum sum of weights such that at least one end point of the edge is selected in $V'$ for any edge in $E$:

$$\min \sum_{v \in V} w_v x_v$$
$$\text{s.t. } x_u + x_v \geq 1, \quad \forall (u,v) \in E,$$
$$x_v \in \{0,1\}, \quad \forall v \in V.$$

In the MIS problem, given an undirected graph $G = (V, E)$, we want to select the largest subset of nodes $V' \subseteq V$ such that no two nodes in the subsets are connected by an edge in $G$:

$$\min - \sum_{v \in V} x_v$$
$$\text{s.t. } x_u + x_v \leq 1, \quad \forall (u,v) \in E,$$
$$x_v \in \{0,1\}, \quad \forall v \in V.$$

In the WA problem, let $I$ denote the set of workers, indexed by $i \in I$, and $J$ denote the set of workloads, indexed by $j \in J$. For each workload $j \in J$, let $A_j \subseteq I$ represent the subset of workers that are eligible to process workload $j$.

Each worker $i \in I$ is associated with an activation cost $c_i \in \mathbb{R}_+$ and a processing capacity $C_i \in \mathbb{R}_+$. Each workload $j \in J$ requires a processing demand of $d_j \in \mathbb{R}_+$. The cost incurred for reserving one unit of capacity on worker $i$ for workload $j$ is denoted by $\gamma_{ij} \in \mathbb{R}_+ \cup \{+\infty\}$, where $\gamma_{ij} = +\infty$ if $i \notin A_j$, indicating that such an allocation is infeasible.

The decision variables are defined as follows: $x_i \in \{0,1\}$ indicates whether worker $i$ is activated ($x_i = 1$) or not ($x_i = 0$), and $y_{ij} \in \mathbb{R}_+$ represents the amount of capacity reserved on worker $i$ for workload $j$. The objective is to minimize the total cost of worker activation and capacity reservation, subject to capacity constraints, assignment feasibility, and robustness against single-worker failures. The resulting mathematical model is formulated as follows:

$$\min \quad \sum_{i \in I} c_i x_i + \sum_{i \in I} \sum_{j \in J} \gamma_{ij} y_{ij}$$
$$\text{s.t.} \quad y_{ij} \leq C_i x_i, \qquad\qquad \forall i \in I, \forall j \in J$$
$$\sum_{j \in J} y_{ij} \leq C_i, \qquad\qquad \forall i \in I$$
$$\sum_{k \in I \setminus \{i\}} y_{kj} \geq d_j, \qquad\qquad \forall j \in J, \forall i \in I$$
$$y_{ij} = 0, \qquad\qquad \forall j \in J, \forall i \notin A_j$$
$$x_i \in \{0,1\}, \qquad\qquad \forall i \in I$$
$$y_{ij} \geq 0, \qquad\qquad \forall i \in I, \forall j \in J$$

## C.4 PROBLEM TEXT DESCRIPTIONS

Due to space limitations, we only present the JSON-formatted representations of the textual descriptions of problems. For the specific textual content of some problems, we employed large language models (LLMs) for generation. Concretely, we supplied the LLM with detailed descriptions of this category of problems as well as the relevant mathematical models, and defined the output format to generate the content we required. The complete content can be found below:

Listing 1: Example JSON Description for the CA Problem

```
{
  "task_description": "The Combinatorial Auction (CA) task is a ...",
  "variable_type": {
    "x": {
      "type": "binary",
```

```
6        "description": "The variable x_i indicates whether bid i ...",
7        "index": "x_i",
8        "range": "[0, 1]",
9        "constraints": "For each bid i, x_i is a binary ..."
10     }
11   },
12   "constraint_type": {
13     "Item Allocation Constraint": {
14        "type": "linear, inequality, leq, SetPacking",
15        "description": "Constraint c_j ensuring that each item j ...",
16        "index": "c_j (j <= n_items )",
17        "expression": "sum(x_i for i in bids) <= 1 ...",
18        "constraints": "For each item j, the sum of x_i ..."
19     },
20     "Substitutable Bids Constraint":{
21        ...
22     }
23   },
24   "edges": [
25     {"source": "0", "target": "0"},
26     {"source": "0", "target": "1"},
27     {"source": "0", "target": "2"}
28   ]
29}
```

In addition, for the pre-trained language model, we selected T5-base. Considering efficiency, we also experimented with some other lightweight language models, such as Sentence-BERT Reimers & Gurevych (2019). However, we did not find any significant differences. The more influential factor turned out to be the training process. Therefore, we did not conduct further attempts in this regard.

### C.5 BASELINES INTRODUCTION

**Predict-and-Search**    Predict-and-Search (PS) is a ML-based framework, which consists of two parts, predict and search. During the prediction phase, PS learns the Bernoulli distribution of the solution values through a GNN network, and get the probability $p_\theta(x_i \mid \mathcal{I})$ for each binary variable, where $\mathcal{I}$ is the MILP instance. Subsequently, based on the marginal probabilities, PS greedily selects $k_0$ binary variables $\mathcal{X}_0$ and fixes them to 0, and selects $k_1$ binary variables $\mathcal{X}_1$ and fixes them to 1. In the searching phase, through the hyperparameter $\Delta$, which determines the range of the neighborhood search, PS employs a traditional solver, such as SCIP or Gurobi, to search in the neighborhood $\mathcal{B}(\mathcal{X}_0, \mathcal{X}_1, \Delta) = \{x : \sum_{x_i \in \mathcal{X}_0} x_i + \sum_{x_i \in \mathcal{X}_1}(1 - x_i) \leq \Delta\}$, where $\mathcal{B}(\mathcal{X}_0, \mathcal{X}_1, \Delta)$ is called the trust-region. The above neiborhood search process can be formulated as the following MILP problem:

$$\min \mathbf{c}^\top \mathbf{x} \quad \text{s.t.} \quad \mathbf{A}\mathbf{x} \leq \mathbf{b}, \quad x \in \mathcal{D} \cap \mathcal{B}(\mathcal{X}_0, \mathcal{X}_1, \Delta), \tag{19}$$

where $\mathcal{D}$ denotes the original feasible region of the MILP. Note that, when $\Delta = 0$, PS degenerates into Neural Diving Nair et al. (2020) (ND)(without selectiveNet).

## D    ALGORITHM

In this section, we present the pseudo-code of the TCP module, which is to identify and select critical tight constraints for a class of MILP, in the algorithm 1. In addition to selecting based on fixed strength $\rho$, to maintain stability and effectiveness during training, we strive to keep critical tight constraints at a relatively balanced proportion. Specifically, we uniformly set $n_{min} = 5$ and $[\beta_1, \beta_2] = [0.1, 0.8]$. It should be noted, however, that this does not require a strict configuration.

## E    COMPLEXITY ANALYSIS

**GCNN**    For the GCNN proposed by Gasse et al. (2019), the neural network comprises a single graph convolution layer followed by an output layer. The graph convolution consists of two half-convolutions (variable-to-constraint and constraint-to-variable), where each edge is processed by

---

**Algorithm 1** TCP Module: identify and select critical tight constraints

---

**Input:** Set of constraint categories $\mathcal{C}$, Fixed constraint strength $\{\rho_i\}_{i=0}^{|\mathcal{C}|}$, JUDGE($\cdot$) in Algorithm 2
**Output:** Selected constraint categories $\mathcal{C}^*$.
1: Sort $\mathcal{C}$ using $\{\rho_i\}_{i=0}^{|\mathcal{C}|}$, $\mathcal{C}_{sorted} = Sort(c)$ .
2: Initialize: $\mathcal{C}^* = \{\}$, $tight\_count = 0$, $total\_count = 0$
3: **for** $\mathcal{C}_i$ in $\mathcal{C}_{sorted}$ **do**
4:     $\hat{\mathcal{C}} = \mathcal{C}^* \cup \{C_i\}$
5:     **if** JUDGE($\hat{\mathcal{C}}$) **then**
6:        $\mathcal{C}^* = \hat{\mathcal{C}}$
7:     **else**
8:        **return** $\mathcal{C}^*$
9:     **end if**
10: **end for**
11: **return** {} {No subset found that meets the criteria}

---

**Algorithm 2** JUDGE

---

**Input:** Constraint set $\mathcal{C}$, Minimum number $n_{min}$, Acceptable proportion $[\beta_1, \beta_2]$.
1: Initialize: $tight\_count = 0$, $total\_count = 0$
2: **for** $C_i$ in $\mathcal{C}$ **do**
3:     $tight\_count = tight\_count + TightConsNum(C_i)$
4:     $total\_count = total\_count + ConstraintNum(C_i)$
5:     **if** $tight\_count \geq n_{min}$ and $\beta_1 \leq (tight\_count/total\_count) \leq \beta_2$ **then**
6:        **return** TRUE
7:     **else**
8:        **return** FALSE
9:     **end if**
10: **end for**

---

two-layer perceptrons, incurring a complexity of $\mathcal{O}(|E| \cdot d^2)$. Updates to constraint and variable nodes contribute $\mathcal{O}(|C| \cdot d^2 + |V| \cdot d^2)$, but since $|E| \gg |C|, |V|$, this is dominated by $\mathcal{O}(|E| \cdot d^2)$. The output layer applies a two-layer perceptron to variable nodes, with complexity $\mathcal{O}(|V| \cdot d^2)$, which is subsumed by the graph convolution as $|V| < |E|$. Thus, the total computational complexity is $\mathcal{O}(|E| \cdot d^2)$.

**Our model** Compared to previous GNN models, the time complexity of our model primarily increases due to the addition of higher GNN and Inter-Layer MP. We assume that the abstract graph contains $N$ category nodes, while the instance graph consists of $n$ instance nodes, On average, each category node corresponds to $d_c$ instance nodes, and the feature dimension is $d$. Since the number of abstract graph nodes $N \ll n$, higer GNN's computational cost is negligible. For Inter-Layer MP, according to Eq. (4)-(7), the computation complexity is $\mathcal{O}(n \cdot d^2 + N \cdot d^2 + n \cdot d + N \cdot d^2)$. Because of $d \gg d_c$, the complexity is approximately $\mathcal{O}(n \cdot d^2)$ and the bottleneck of this process is the Cross-Attention operation. Therefore, after considering the underlying GCNN, the total complexity is $\mathcal{O}(n \cdot d^2 + |E| \cdot d^2)$.

## F    DETAILS ON BIPARTITE GRAPH REPRESENTATION

In this paper, we use basically the same features as those in PS Han et al. (2023) to characterize the instance bipartite graph. Specifically, we also add position embedding to the constraints. The specific features are shown in the Table 7.

Table 7: Features in embedded bipartite representations.

| | # features. | name | description |
|---|---|---|---|
| Variable | 1 | obj | normalized coefficient of variables in the objective function |
| | 1 | v_coeff | average coefficient of the variable in all constraints |
| | 1 | Nv_coeff | degree of variable node in the bipartite representation |
| | 1 | max_coeff | maximum value among all coefficients of the variable |
| | 1 | min_coeff | minimum value among all coefficients of the variable |
| | 1 | int | binary representation to show if the variable is an integer variable |
| | 12 | pos_emb | binary encoding of the order of appearance for each variable among all variables. |
| Constraint | 1 | c_coeff | average of all coefficients in the constraint |
| | 1 | Nc_coeff | degree of constraint nodes in the bipartite representation |
| | 1 | rhs | right-hand-side value of the constraint |
| | 1 | sense | the sense of the constraint |
| Edge | 1 | coeff | coefficient of variables in constraints |

## G  HYPERPARAMETERS

### G.1  TRAINING PARAMETER

All evalutions are performed under the same configurations. For the training of the model, we uniformly set the learning rate as $10^{-3}$ with Adam optimizer (Kingma & Ba, 2014), the batch size as $4$, and the max epoch as $150$, although models typically achieve convergence within 50 to 100 epochs. Regarding the parameters of the Focal Loss, we keep them consistent with the conclusions in Lin et al. (2017) and set $\alpha = 0.75, \gamma = 2$ for MIS and MVC. And we set the balancing weight of the final loss $\lambda = 0.4$ in Eq. (14) for all benchmarks. For the pre-trained language model used in text embedding, considering efficiency, we choose T5-base (Raffel et al., 2020).

### G.2  NEIGHBORHOOD PARAMETERS

We report the neighborhood Parameters for all baselines with Gurobi in Table 8.

Table 8: Hyperparameters $(k_0, k_1, \Delta)$ for PS and ConPaS; $(k_0, k_1, \Delta, k_c, \Delta_c)$ for ours.

| Benchmark | CA | MIS | MVC | WA |
|---|---|---|---|---|
| PS | (1200, 0, 20) | (1200, 600, 20) | (600, 100, 10) | (0, 550, 10) |
| ConPas | (1200, 0, 20) | (1200, 600, 20) | (600, 200, 20) | (0, 500, 5) |
| Ours | (1000, 0, 10, 80, 15) | (600, 600, 5, 800, 1) | (600,200,20,1000, 1) | (0, 550, 10, 10000, 1) |

## H  ADDITIONAL EXPERIMENTAL RESULTS

### H.1  MOTIVATION EXPERIMENTS

We conducted motivational experiments on toy datasets, CA and WA. Specifically, we only fix the constraints without performing any other operations. Because a full enumeration of all possible combinations would be computationally infeasible, we chose to test 20 combinations selected based on our empirical observations. To illustrate, in the case of the CA problem, we noticed that fixing constraints with fewer variables generally results in a faster solution. Consequently, to construct our test sets, we began by sampling constraints that involved the fewest and the most variables and

then randomly created combinations to be fixed during the solve. We show the results of 8 of these combinations in the table 9.

Table 9: the results of 10 of these combinations of CA.

|  | $c_1$ | $c_2$ | $c_3$ | $c_4$ | $c_5$ | $c_7$ | $c_8$ | $c_9$ |
|---|---|---|---|---|---|---|---|---|
| Time | **1.85** | 332.72 | **465.74** | 15.49 | 98.44 | 370.51 | 51.57 | 84.01 |
| Avg var num | **3.0** | 21.2 | **34.2** | 9.2 | 14.8 | 24.1 | 12.1 | 14.8 |

The Table 1 shows the average solving time across 30 instances under different operations. Here, "Original" denotes the initial solving time. Based on our experience and findings, we evaluate different combinations of $n$ distinct constraints, identifying the fastest solution as the "Best Fixing" and the slowest as the "Worst Fixing". For CA_easy, $n = 10$, and for WA, $n = 3000$. And for the CA_easy dataset used here, there are on average 600 constraints and 1,500 variables.

## H.2 COMPARSION WITH GUROBI

In this section, we supplement the experimental results of baseline with Gurobi.

**Main Evalution** In Figure 5, we have shown the relative primal gap as a function of runtime for CA, MIS, and MVC. As can be seen from the Figure 5, among the ML-based methods using Gurobi across multiple benchmarks, our method is still able to achieve the best solution quality and convergence speed. For the two datasets, MIS and MVC, since the solutions obtained by our method are better than those obtained by Gurobi after 3600 seconds of solving, the solutions obtained by our method are taken as the Best Known Solutions (BKS). In this regard, for the primal gap, we have truncated it to a minimum of $1 \times 10^{-6}$ in the figure. Since the WA instances solved by Gurobi can rapidly reach objective values comparable to the BKS (Best Known Solution), the figure in Figure 5 fails to effectively illustrate performance disparities. Therefore, we deliberately omit the presentation of Gurobi-based WA results in this context.

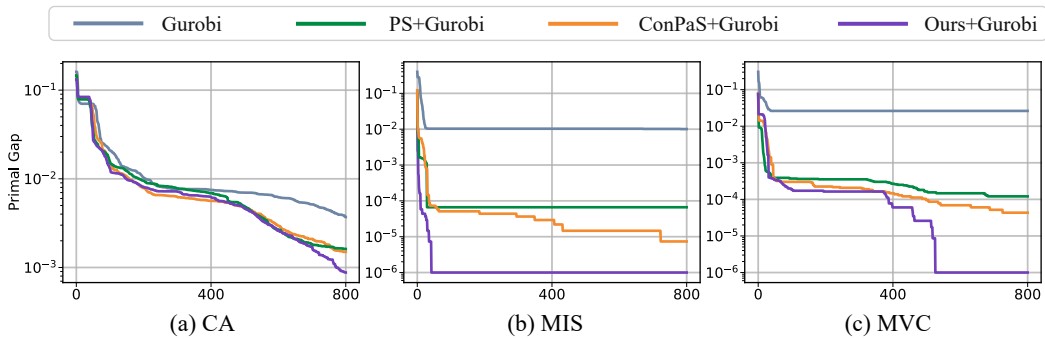

Figure 5: The relative primal gap (the lower the better) based Gurobi as a function of runtime, averaged over 100 test instances, within 800s time limits.

**Real-world Datasets** Here, we report the Gurobi results for the real-world dataset MMCN in the Figure 6. As shown as figure, our method also outperforms all the baselines on the real-world dataset, which demonstrates that our method exhibits excellent performance when using different solvers.

## H.3 HYPERPAPAMETERS ANALYSIS

In this section, we explore the effect of the neighborhood parameter hyperparameters on CA dataset based Gurobi. $(k_0, k_1, \Delta)$ have been studied numerous times in previous works. Here, we focus on studying $(k_c, \Delta_c)$ which is a crucial hyperparameter that has a significant impact in our work. The results are shown on Table 10. We observe that as the value of $k_c$ increases, the error rate of neural network predictions also increases, leading to a degradation in solution quality. However, the convergence speed shows a certain degree of improvement.

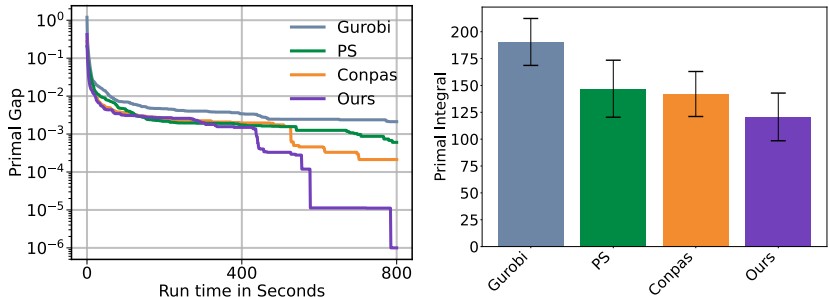

Figure 6: The left: the relative primal gap (the lower the better) based Gurobi as a function of runtime, averaged over 100 test instances, within 800s time limits on the real-world datasets. The right: the primal-dual integral (PDI) at 800s runtime cutoff based Gurobi. The error bars represent the standard deviation.

Table 10: Performance of different fixed constraints.

| $(k_c, \Delta_c)$ | Gurobi | | SCIP | |
|---|---|---|---|---|
| | $\text{gap}_{\text{abs}} \downarrow$ | PDI $\downarrow$ | $\text{gap}_{\text{abs}} \downarrow$ | PDI $\downarrow$ |
| (10, 0) | 492.78 | 74.88 | 3945.76 | 111.03 |
| (20, 0) | 109.57 | 73.07 | **3401.63** | 109.41 |
| (40, 5) | 375.73 | 73.56 | 4270.87 | 114.64 |
| (50, 5) | 228.69 | 73.52 | 4054.70 | 108.69 |
| **(80, 15)** | **104.72** | **73.02** | 3485.82 | **108.23** |
| (100, 20) | 161.81 | 73.41 | 3673.37 | 109.11 |
| (120, 25) | 217.94 | 73.98 | 4535.83 | 110.73 |

Additionally, taking the CA instances as an example, Table 11 demonstrates how solver performance degrades with different fractions of incorrectly fixed constraints. "N/A" denotes problem infeasibility and we randomly choose three CA instances.

Table 11: the results of solver performance degrades with different fractions of incorrectly fixed constraints

| Incorrectly Fixed | CA12 | | CA20 | | CA25 | |
|---|---|---|---|---|---|---|
| | Time(s) | Objective | Time(s) | Objective | Time(s) | Objective |
| 0% | 154.47 | 24543.48 | 163.58 | 25055.95 | 79.31 | 24181.19 |
| 2% | 137.58 | 24325.03 | 199.99 | 24810.55 | 107.05 | 23564.07 |
| 5% | 37.80 | 23358.41 | 11.14 | 23891.40 | 108.18 | 23445.82 |
| 10% | 4.96 | 22160.16 | 23.93 | 23516.66 | N/A | N/A |
| 20% | N/A | N/A | N/A | N/A | N/A | N/A |
| 30% | N/A | N/A | N/A | N/A | N/A | N/A |

In this experiment, we begin with a fixed set of correctly identified tight constraints (5% of tight constraints). We then introduce varying proportions of non-tight (i.e., incorrectly fixed) constraints to this set. The resulting degradation in solver performance is then measured by the solution time and the final objective value.

## H.4 PRESOLVE ABLATION STUDY

To quantify the interaction between the learned model reduction and the solver's presolve, we conduct a presolve ablation study on the MVC and CA datasets using Gurobi. Unless otherwise stated, Gurobi is run with presolve disabled; we then explicitly enable presolve on top of each reduction variant.

Table 12: Presolve ablation on MVC and CA. Lower $\text{gap}_{\text{abs}}$ and PDI indicate better performance.

| Method | MVC $\text{gap}_{\text{abs}}$ | MVC PDI | CA $\text{gap}_{\text{abs}}$ | CA PDI |
|---|---|---|---|---|
| Gurobi | 8.38 | 68.02 | 565.61 | 76.78 |
| + presolve | 7.44 | 72.31 | 477.51 | 76.29 |
| PS | 0.28 | 7.39 | 420.47 | 78.20 |
| + presolve | 0.20 | 8.28 | 210.19 | 73.78 |
| Ours | 0.00 | 6.26 | 410.58 | 77.29 |
| + presolve | 0.00 | 4.56 | 104.72 | 73.02 |

Table 12 reports the average absolute optimality gap ($\text{gap}_{\text{abs}}$) and PDI on both datasets for: vanilla Gurobi, Gurobi with presolve, the classical reduction baseline (PS), and our learned reduction, each with and without presolve.

From Table 12, we make three observations:

1. Combining our learned reduction with presolve yields the best overall performance on both datasets, indicating that the two mechanisms are complementary rather than conflicting.

2. Under the same solver configuration, the learned reduction consistently achieves better solution quality than the presolve method.

3. For vanilla Gurobi as well as for both reduction methods, enabling presolve consistently improves performance, confirming that presolve remains beneficial even in the presence of strong model reductions.

These results highlight the actual contribution of the learned reduction and show that it can be effectively combined with standard presolve techniques to further improve solver performance.

## H.5 ANALYSIS OF MULTIMODAL REPRESENTATION

In Figure 4, there is little discernible gain precisely at the Time Limit. However, before reaching the time limit, our multimodal representation consistently demonstrates significant benefits. For example, at the 400-second mark, our method achieves a 12.0% improvement in the primal gap. In Table 13, we show the loss and the prediction accuracy of constraints for our multimodal model.

Table 13: the training metrics of our multimodal model. $ACC@K$ means the accuracy of constraints when K% of tight constraints need to be fixed.

| | loss | $ACC_{var}$ | $ACC@10$ | $ACC@20$ | $ACC@50$ | $ACC@100$ |
|---|---|---|---|---|---|---|
| ours | 1246.3 | 99.31% | 93.13% | 84.84% | 66.41% | 64.30% |
| GNN | 1263.8 | 98.80% | 90.37% | 80.62% | 66.37% | 64.31% |

Moreover, in the main experiments, we use the T5-base model as the text encoder. To assess the sensitivity of our approach to the choice of pretrained language model (PLM), we further replace T5-base with the stronger OpenAI `text-embedding-3-large` model (called OpenAI Embed).[2] The comparison is summarized in Table 14.

As shown in Table 14, switching from T5-base to OpenAI `text-embedding-3-large` leads to only marginal improvements in loss, absolute gap, and PDI. This suggests that the overall performance is not particularly sensitive to the specific PLM. A key reason is that, compared with the relatively coarse sentence-level embeddings produced by PLMs, the MLP used to align PLM embeddings with the GNN representations plays a more critical role, effectively mitigating the influence of the PLM choice itself.

---

[2]We keep the intermediate feature dimension unchanged and leave all other settings identical to those in the original Gurobi-based experiments.

Table 14: Effect of the pretrained language model on the Gurobi-based experiments in CA dataset.

| PLM | Loss | $\text{gap}_{\text{abs}}$ | PDI |
|---|---|---|---|
| T5-base | 1246.3 | 104.72 | 73.02 |
| OpenAI Embed | 1232.1 | 103.33 | 72.96 |

## H.6 TRAINING TIME CONSUMPTION

In this section, since we have employed a more complex model, we report the training duration per epoch of our model compared to that of the naive Graph Neural Network (GNN) on multiple datasets.

Table 15: Per-epoch training time (in seconds) for PS (naive GNN) and our model on different datasets.

| Dataset | PS (GNN) | Ours |
|---|---|---|
| CA | 9.7 | 23.1 |
| MIS | 66.5 | 148.4 |
| MVC | 60.2 | 122.6 |
| WA | 242.4 | 358.9 |

## H.7 DISCUSSION ABOUT ASSUMPTION (10)

To assess the practical validity of the local decoupling assumption, we conduct an additional empirical study on the same MILP benchmarks used in the main paper, including CA, WA, and the real-world dataset MMCN.

**Empirical evidence on realistic datasets.** For each instance, we

1. group constraints into prototypical types (Set Packing, Set Partitioning, Knapsack, General Linear);
2. for every constraint, compare (i) the distribution of its activities and (ii) the marginal distributions of its incident variables under solutions of the *full model*, and solutions of a *type-only local model* that retains only constraints of that type, as implied by the local decoupling assumption.

We use the Wasserstein distance (WD) on constraint activities and the mean absolute error on variable marginals (VarErr) to measure the distributional discrepancy between the two solutions. A constraint is counted as *satisfying* the local decoupling assumption under a given metric if WD $< 0.25$ (for activities) or VarErr $< 0.15$ (for marginals).

The averaged statistics over 40 instances are summarized in Table 16.

**Key observations.** We highlight several observations from Table 16:

- On CA, a substantial fraction of Set Packing constraints (40.1% by WD and 58.3% by VarErr) is well approximated by the corresponding type-only local model, even under this deliberately strict setting.
- On MMCN2, the highly structured types that our theory focuses on behave particularly well:
  - all Set Packing constraints are perfectly matched in variable marginals (100.0% satisfying VarErr with a very small average error);
  - most General Linear constraints exhibit solution distributions that closely approximate those of the global model.

These results indicate that, in realistic instances, a large and structured subset of constraints— precisely the prototypical types emphasized by our theoretical analysis—satisfies the local decoupling assumption to a meaningful extent.

Table 16: Empirical validation of the local decoupling assumption on realistic MILP benchmarks. "Satisfied (WD)" and "Satisfied (Var)" denote the fraction of constraints whose activity distributions and variable marginals, respectively, are well approximated by the corresponding type-only local models.

| Dataset | Constraint Type | Satisfied (WD) | Satisfied (Var) | Avg WD | Avg VarErr |
|---------|-----------------|----------------|-----------------|--------|------------|
| CA | Set Packing | 40.1% | 58.3% | 0.334 | 0.148 |
| MMCN | General Linear | 25.7% | 87.3% | 91.096 | 13.330 |
| MMCN | Set Packing | 18.5% | 100.0% | 0.758 | 0.063 |
| WA | General Linear | 49.9% | 11.2% | 0.379 | 0.360 |
| WA | Mixed Binary | 8.2% | 7.4% | 0.685 | 0.328 |

Moreover, in practice we observe that the constraints selected to be fixed by our reduction procedure are exactly those exhibiting a smaller gap between (i) the distributions implied by the local decoupling assumption and (ii) the empirical distributions estimated from data. This is consistent with our analysis that constraints better satisfying the assumption are more suitable to be fixed. For datasets such as WA, where these gaps are generally larger and constraints appear more strongly coupled, the performance gains brought by our method are indeed smaller than on other datasets, further reinforcing the alignment between theory and empirical behavior.

**Role of the assumption.** The local decoupling assumption is used as a *local analytic device* to derive the entropy-based ranking over constraint types. It is not required for the feasibility or correctness of the reduction procedure itself, but rather serves to motivate why certain prototypical types (e.g., Set Packing / Set Partitioning) are expected to be more informative. The overall method remains heuristic in nature, and its effectiveness is ultimately validated by empirical evidence such as that presented above.

# I   LIMITATION AND FUTURE WORK

This paper proposes a constraint-based model reduction method by reducing both variables and constraints. There are two challenges for constraint reduction: 1) which inequality constraints are critical, and 2) how to predict these critical constraints efficiently. Specifically, we introduce the concept of critical tight constraints and develop a heuristic algorithm to identify them. To address the challenges of representing variables and constraints, we leverage an abstract model of the MILP problem to enable multi-modal representations. However, a limitation of our approach is its reliance on prior knowledge of the specific MILP formulation. Furthermore, the heuristic search for critical tight constraints may not always be sufficient. In future work, we aim to develop a more general and principled method for identifying instance-level critical tight constraints.

