# OpenReview forum: "Constraint Matters: Multi-Modal Representation for Reducing Mixed-Integer Linear programming"
_ICLR.cc/2026/Conference — ICLR 2026 Poster_

### Official Review · Reviewer_3DQK · 2025-10-30

**Soundness:** 2
**Presentation:** 3
**Contribution:** 2
**Rating:** 4
**Confidence:** 4

**Summary:**

This manuscript proposes a method that learns to predict tight constraints for Mixed-Integer Linear Programs, which are then used to construct a reduced sub-problem. The goal is to enable faster solution search compared to the original MILP. The paper provides both theoretical and empirical evidence supporting the proposed method. However, the theoretical results are somewhat trivial and rely on a strong assumption, while the baselines considered for comparison are insufficient.

**Strengths:**

1. The manuscript is well-written with a clear and logical flow.
2. The idea of constraint reduction is well-motivated, and it holds potential for practical impact.
3. Empirical results demonstrate the advantages of the proposed method over the considered baselines.

**Weaknesses:**

1. **Confusing definitions**:
    - In Definition 1, $C_i$ is defined as a constraint type with unspecified parameters a,b and c. However, the phrase "C_i(x) is satisfied" is unclear. How should the satisfaction of a constraint type be interpreted? Clarifying this would improve understanding.
    - In Definition 2, the term "fixing" appears early in the description of properties of the problem. Isn’t "fixing" typically an action performed during the reduction step? Could this be clarified?
2. **Strong assumption**: The local decoupling assumption is quite strong and its applicability in practice remains unclear. It would be useful to provide statistics or empirical evidence demonstrating the proportion of constraints that satisfy this assumption in realistic scenarios. This would help assess the gap between theory and practice.
3. **The main Theorem 4 is trivial and meaningless**: The main result in Theorem 4, which suggests that fixing tight constraints leads to a sub-problem containing the optimal solution, is trivial and lacks meaningful insight for the learning task, considering there is no guarantee that the model will accurately predict all tight constraints.
4. **Inadequate baselines**: The authors mention prior works that also learn tight constraints for model reduction, yet fail to compare their method against these methods. I recommend incorporating at least one of these works as a baseline to enhance the comparison and demonstrate the advantages of the proposed method.

**Questions:**

1. The tight constraints are defined based on the optimal solution. What happens if there are multiple optimal solutions? How does the proposed approach handle such cases? Also, what if obtaining the optimal solution is computationally expensive?
2. An IP dataset is used in both of the considered learning baselines but is excluded from the experiments in this work. Could the authors clarify the reasoning behind this exclusion?

---

> ### Author Response · Authors · 2025-11-19
>
> We are sincerely grateful to the reviewer for their thoughtful, detailed, and incredibly constructive review.
>
> The reviewer's suggestions for strengthening the paper were so clear and insightful that they spurred us to conduct several new, targeted experiments, which we believe have substantially improved our work. We are pleased to present our detailed responses below, addressing each weakness and question point-by-point. We hope our responses, supported by new data, fully clarify the points raised.
> > ###  W1. Confusing definitions
>
> > In Definition 1, C_i is defined as a constraint type with unspecified parameters a,b and c. However, the phrase "C_i(x) is satisfied" is unclear. How should the satisfaction of a constraint type be interpreted? Clarifying this would improve understanding.
>
> A1.1: We apologize for the unclear description. The specific definition of “the satisfaction of a constraint type” is as follows: for a set of variable values $x$ corresponding to the $i$-th constraint category (denoted as $C_i$), if all constraint conditions within this category are satisfied by $x$ without any violations, we refer to this state as “satisfied.”
>
> > In Definition 2, the term "fixing" appears early in the description of properties of the problem. Isn’t "fixing" typically an action performed during the reduction step? Could this be clarified?
>
> A1.2: We apologize again for the unclear description. Fixing a constraint refers to transforming an inequality constraint $a x_i \leq b_i$ into an equality constraint $a x_i = b_i.$
>
> The specific meaning of Definition 2 is as follows: during the constraint reduction process, all tight inequality constraints in the $i$-th constraint category (denoted as $C_i$) are reconstructed into equality constraints. The resulting $i$-th constraint category after this operation is defined as $\hat{C}_i.$
>
> These unclear descriptions will be clarified in the paper, and the definition of “fixing” will be explicitly specified in Section 3.
>
>
> > ###  W2. The local decoupling assumption is quite strong, and its applicability in practice remains unclear. It would be useful to provide statistics or empirical evidence demonstrating the proportion of constraints that satisfy this assumption in realistic scenarios. This would help assess the gap between theory and practice.
>
> A2：For Assumption 1, we intend to show that the method admits favorable theoretical properties when the MILP exhibits certain structural characteristics. Without these characteristics, our proposed method remains effective —experiments confirm strong performance on problems such as CA, MIS, WA, and MVC, all of which lack coupling relationships. Intuitively, reducing the solution space increases information gain and decreases branch-and-bound traversal, thereby accelerating solving. This assumption simply facilitates the theoretical analysis used to formalize this intuition and does not affect the method itself.
>
> To avoid potential misunderstandings regarding Assumption 1, we will reorganize Proposition 3 by directly incorporating Assumption 1. Under this assumption, we can derive $\Delta H_{C_i} = -\log \rho$, which theoretically illustrates that solution space reduction leads to information gain.
>
> The revised Proposition 3 is as follows:
>
> ----
>
> Let $P(X_{C_i} | I)$ represent the probability distribution of the values taken by the variables acted upon by the prototypical constraint $C_{i}$. Assume that
> $$
>     P(X_{C_i} | I) = P(X_{C_i} | C_i, \mathcal{U}) \approx P(X_{C_i} | C_i),
> $$
> where $\mathcal{U}$ denotes the influence noise of other > constraint types in the instance $I$ on the $X_{C_i}$.
> Therefore, we have the local information gain $\Delta H_{C_i}$ of fixing $C_i$ as below:
> $$
> \Delta H_{C_i} = -\log \rho.
> $$
> We interpret the feasible region of a MILP as a space of uncertainty, and fixing constraints can be viewed as a process of entropy reduction. We have that a lower $\rho$ value implies a larger reduction in uncertainty, which in turn effectively shrinks the search space during solving and accelerates the process—this partly demonstrates the validity of our heuristic method based on $\rho$.
>
> ----
>
> In addition, we will add a **footnote** to Proposition 3 stating that experimental results demonstrate the effectiveness of our method even when the problem scenario does not satisfy the local decoupling assumption.
>
> > ###  W3. The main Theorem 4 is trivial and meaningless
>
> A3: It should be clarified that Theorem 4 states that if the neural network’s prediction is correct, both optimality and feasibility are guaranteed. However, if the prediction is inaccurate, suboptimality may arise. Therefore, we introduce a constraint search mechanism (Formula 12) to mitigate the issues caused by such prediction errors.

---

> ### Author Response · Authors · 2025-11-19
>
> > ###  W4. Inadequate baselines: The authors mention prior works that also learn tight constraints for model reduction, yet fail to compare their method against these methods. I recommend incorporating at least one of these works as a baseline to enhance the comparison and demonstrate the advantages of the proposed method.
>
> A4: To further validate the performance advantages of our method, we conducted additional comparative experiments in which a full-time constraint-tightness prediction method (with fixed settings, referred to as **“complete fixing”**) served as the control. The relevant results are shown in the table below.
>
> | method          | $gap_{abs}$ | PDI   |
> | --------------- | ------------- | ----- |
> | Gurobi          | 477.51        | 76.29 |
> | PS              | 210.19        | 73.78 |
> | complete fixing | 183.27        | 73.86 |
> | ours            | 104.72        | 73.02 |
>
> It should be clarified that existing tight-constraint reduction methods lack generalization ability and are designed for specific application scenarios. Therefore, a direct performance comparison across different methods would be unfair—this is the primary reason we did not include them as baselines.
>
> Work [2] (extended from [1]) relies on costly sampling to construct active-constraint candidates and uses instance-level “instance-to-set” prediction. However, its candidate set grows exponentially with problem scale and parameter ranges, making it unsuitable for scenarios with substantial parameter fluctuations or dynamic changes in problem size. In contrast, while such methods are limited to fast-solving scenarios with mild parameter variations, our approach generalizes well under both parameter and scale changes, pioneering constraint reduction for general MILP settings.
>
> [1] Bertsimas, Dimitris, and Bartolomeo Stellato. “Online mixed-integer optimization in milliseconds.” *INFORMS Journal on Computing* 34.4 (2022): 2229–2248.
>
> [2] Li, Yixuan, et al. “Fast and interpretable mixed-integer linear program solving by learning model reduction.” *Proceedings of the AAAI Conference on Artificial Intelligence*. Vol. 39. No. 25. 2025.
>
> > ### Question 1: The tight constraints are defined based on the optimal solution. What happens if there are multiple optimal solutions? How does the proposed approach handle such cases? Also, what if obtaining the optimal solution is computationally expensive?
>
> To clarify, each feasible solution corresponds to a set of tight constraints. However, what we aim to learn is the (sub-)optimal solution. Regarding your question about the “existence of multiple optimal solutions,” we acknowledge that this scenario may indeed occur, but it does not hinder the effectiveness of our method. We do not need to consider all optimal solutions; selecting any one solution as the label is sufficient, as long as it is consistent with its corresponding tight constraints. The subsequent neighborhood search mechanism (Formula 12) then ensures consistency between variables and constraints.
>
> Concerning the concern about the “high computational cost of optimal solutions,” it should be emphasized that computing optimal solutions is an offline preprocessing step. Most learning-based methods likewise rely on optimal (or near-optimal) solutions as supervision. Moreover, in practice, a strictly optimal solution is not always necessary—a high-quality solution can serve as an effective training label. During the later search phase, the method still has the potential to converge to the optimal solution.
>
> In our experiments, we collected solutions from Gurobi with a 1800s time limit as training labels (denoted as “ours w/ subopt”). The results for CA are shown in the table below.
>
> | method         | $gap_{abs}$ | PDI   |
> | -------------- | ------------- | ----- |
> | Gurobi         | 477.51        | 76.29 |
> | ours w/ subopt | 106.37        | 73.09 |
> | ours           | 104.72        | 73.02 |
>
> It can be observed that the results are barely affected. This is because between 1800s and 3600s, the solver primarily improves the dual bound, while the primal bound remains essentially unchanged.
>
>
> > ###  Question 2. An IP dataset is used in both of the considered learning baselines but is excluded from the experiments in this work. Could the authors clarify the reasoning behind this exclusion?
>
> The IP dataset corresponds to a relatively special type of problem scenario: the vast majority of its constraints are equality constraints, while tight inequality constraints constitute only a very small proportion.
>
>  For variable-based reduction methods, this characteristic does not affect their performance. However, such scenarios are incompatible with our constraint-centric method, whose core innovation lies in constraint reduction. This is the primary reason why we did not include this dataset in the paper’s experiments, as stated in line 882.

---

> > ### Comment · Reviewer_3DQK · 2025-11-22
> >
> > Thank you to the authors for the detailed rebuttal. Most of my concerns have been addressed, except for one regarding the theoretical aspects. Specifically, the gap between the local decoupling assumption and the practical data remains unclear. Why not directly present the proportion of constraints satisfied by this assumption in realistic scenarios, such as the datasets considered? Highlighting this gap would be crucial for providing a more comprehensive understanding.

---

> > > ### Author Response · Authors · 2025-11-24
> > >
> > > >  **Response to the concern about the local decoupling assumption**
> > >
> > > We appreciate the reviewer’s concern that the local decoupling assumption may look strong, and its practical validity needs to be quantified.
> > >
> > >
> > > **(1) Empirical evidence on realistic datasets.**
> > >
> > > To directly address the reviewer’s request, we conducted an additional study on the same MILP benchmarks as in the paper, including CA, WA, and a real-world dataset MMCN.
> > >
> > > For each instance, we:
> > >
> > > * group constraints into prototypical types (Set Packing, Set Partitioning, Knapsack, General Linear);
> > > * Compare, for every constraint, the distribution of its activities and the marginal distributions of its incident variables under
> > >   (i) solutions of the **full model** and
> > >   (ii) solutions of a **type-only local model** that keeps only constraints of that type, based local decoupling assumption.
> > >
> > > We use the Wasserstein distance (WD) on activities and mean absolute error on variable marginals (VarErr) to measure the distributional discrepancy between the two solutions.. A constraint is counted as “satisfying” local decoupling under a metric if WD < 0.25 (for activities) or VarErr < 0.15 (for marginals).
> > >
> > > The averaged statistics over 40 instances are:
> > >
> > >
> > > | Dataset | Constraint Type | Satisfied (WD) | Satisfied (Var) | Avg WD   | Avg VarErr |
> > > |---------|-----------------|----------------|-----------------|----------|------------|
> > > | CA      | Set Packing     | 40.1%          | 58.3%           | 0.334    | 0.148      |
> > > | MMCN   | General Linear  | 25.7%          | 87.3%           | 91.096   | 13.330     |
> > > | MMCN   | Set Packing     | 18.5%          | 100.0%          | 0.758    | 0.063      |
> > > | WA      | General Linear  | 49.9%          | 11.2%           | 0.379    | 0.360      |
> > > | WA      | Mixed Binary    | 8.2%           | 7.4%            | 0.685    | 0.328      |
> > >
> > >
> > >
> > > **(2) Key observations.**
> > >
> > > * On **CA**, a substantial fraction of Set Packing constraints (40.1% by WD, 58.3% by VarErr) is well approximated by the local type-only model, even under this deliberately strict setting.
> > > * On **MMCN**, the highly structured types that our theory focuses on behave particularly well:
> > >
> > >   * All set Packing constraints are perfectly matched in variable marginals (100% satisfying VarErr with a very small average error).
> > >   * Most General Linear constraints exhibit solution distributions that closely approximate the global solution distribution.
> > >
> > > These results show that, in realistic instances, **a large and structured subset of constraints—precisely the prototypical types emphasized by our theory—does satisfy the local decoupling assumption to a good extent**.
> > >
> > > Interestingly, we observe that, in practice, the constraints selected to be fixed by our method are precisely those exhibiting **smaller gap between the distributions implied by the local decoupling assumption and the empirical distributions observed from data**. This is fully consistent with our analysis that constraints better satisfying the local decoupling assumption are more suitable to be fixed. Moreover, for datasets such as **WA**, where these gaps are generally larger and the constraints are more strongly coupled, the performance gains brought by our method are indeed smaller compared with other datasets. This further demonstrates the consistency between our theoretical findings and empirical observations.
> > >
> > >
> > > **(3) Role of the assumption.**
> > >
> > > Our local decoupling assumption is used as a *local analytic device* to derive the entropy-based ranking over constraint types. It is **not** required for the feasibility or correctness of our reduction procedure, but serves to motivate why certain prototypical types (e.g., set packing/partitioning) are expected to be more informative. The overall method remains heuristic, and its effectiveness is ultimately validated by empirical results.
> > >
> > > We will add this empirical study and the above table as a new subsection (“Empirical validation of the local decoupling assumption”) to make the connection between theory and practice more explicit.
> > >
> > > -------
> > > We hope that our detailed responses and the new empirical evidence have fully addressed your questions and have further clarified the contributions of our paper. We are, of course, very eager to engage in any further discussion and would be delighted to answer any additional questions you may have.
> > >
> > > Thank you again for your invaluable guidance in improving our work.

---

> > > > ### Comment · Reviewer_3DQK · 2025-11-24
> > > >
> > > > I appreciate the authors' detailed response and will revise my score accordingly. I suggest that the authors incorporate the clarifications and experiments mentioned in the rebuttal into the revised manuscript, and clearly emphasize the role of the assumption, as outlined in their response.

---

> > > > > ### Author Response · Authors · 2025-11-26
> > > > >
> > > > > Thank you very much for your considerate follow-up and for revising your score. We will carefully incorporate the clarifications and additional experiments from the rebuttal into the revised manuscript and explicitly emphasize the role of the assumption as you suggested. Your feedback is greatly appreciated and will help us improve the paper.

---

### Official Review · Reviewer_HaxL · 2025-10-30

**Soundness:** 3
**Presentation:** 3
**Contribution:** 4
**Rating:** 4
**Confidence:** 4

**Summary:**

This paper proposes a novel constraint-based model reduction framework for Mixed-Integer Linear Programs (MILPs). The core idea is to identify and fix a subset of "critical" tight constraints to reduce the problem's complexity, complementing the more common variable-fixing approaches. Experiments on standard benchmarks and a real-world dataset show improvements in solution quality and solving time.

**Strengths:**

1. The paper's focus on constraint reduction is a valuable direction. The idea that not all tight constraints are equally valuable for acceleration is intuitive, and providing a data-driven method to identify them is a meaningful contribution to the ML-for-optimization community.
2. The proposed GNN architecture that incorporates both the instance-specific bipartite graph and an abstract graph with textual embeddings is a sophisticated and well-motivated approach. Fusing information from the problem category level is a sensible way to improve generalization and prediction accuracy for constraints.

**Weaknesses:**

1. While the information-theoretic motivation for the TCP heuristic is appealing, its practical implementation relies heavily on the "Local Decoupling Assumption" (Assumption 1). This assumption is a significant simplification, as constraints in MILPs are inherently coupled. The paper acknowledges that this is a heuristic, but the leap from the theoretical to its application for selecting constraints within a specific instance requires further justification. How does the pre-computed ρ for a constraint type reliably indicate the criticality of a particular constraint instance whose variables interact with many other constraints? A more detailed discussion or empirical validation of this link's robustness would strengthen the method's foundation.
2. ​Baseline comparison is lacking in details, since Gurobi already has a lot of presolving techniques used to reduce variables and constraints. A fairer comparison needs to turn these presolving methods off.

**Questions:**

1. What is the setting used in the experiments for the MILP Solver?

---

> ### Author Response · Authors · 2025-11-19
>
> We are sincerely grateful to the reviewer for their thoughtful, detailed, and incredibly constructive review.
>
> The reviewer's suggestions for strengthening the paper were so clear and insightful that they spurred us to conduct several new, targeted experiments, which we believe have substantially improved our work. We are pleased to present our detailed responses below, addressing each weakness and question point-by-point. We hope our responses, supported by new data, fully clarify the points raised.
>
> > ###  W1. ..., How does the pre-computed ρ for a constraint type reliably indicate the criticality of a particular constraint instance whose variables interact with many other constraints? A more detailed discussion or empirical validation of this link's robustness would strengthen the method's foundation.
>
> A1：For Assumption 1, our intention is to show that the method admits favorable theoretical properties when the MILP exhibits certain structural characteristics. Without these characteristics, our proposed method remains effective —experiments confirm strong performance on problems such as CA, MIS, WA, and MVC, all of which lack coupling relationships. Intuitively, reducing the solution space increases information gain and decreases branch-and-bound traversal, thereby accelerating solving. This assumption simply facilitates the theoretical analysis used to formalize this intuition and does not affect the method itself.
>
> To avoid potential misunderstandings regarding Assumption 1, we will reorganize Proposition 3 by directly incorporating Assumption 1. Under this assumption, we can derive $\Delta H_{C_i} = -\log \rho$, which theoretically illustrates that solution space reduction leads to information gain.
>
> The revised Proposition 3 is as follows:
>
> ---
>
> Let $P(X_{C_i} | I)$ represent the probability distribution of the values taken by the variables acted upon by the prototypical constraint $C_{i}$. Assume that
> $$
>     P(X_{C_i} | I) = P(X_{C_i} | C_i, \mathcal{U}) \approx P(X_{C_i} | C_i),
> $$
> where $\mathcal{U}$ denotes the influence noise of other > constraint types in the instance $I$ on the $X_{C_i}$.
> Therefore, we have the local information gain $\Delta H_{C_i}$ of fixing $C_i$ as below:
> $$
> \Delta H_{C_i} = -\log \rho.
> $$
> We interpret the feasible region of a MILP as a space of uncertainty, and fixing constraints can be viewed as a process of entropy reduction. We have that a lower $\rho$ value implies a larger reduction in uncertainty, which in turn effectively shrinks the search space during solving and accelerates the process—this partly demonstrates the validity of our heuristic method based on $\rho$.
>
> ---
> In addition, we will add a **footnote** to Proposition 3 stating that experimental results demonstrate the effectiveness of our method even when the problem scenario does not satisfy the local decoupling assumption.
>
>
> > ###  W2. ​Baseline comparison is lacking in details, since Gurobi already has a lot of presolving techniques used to reduce variables and constraints. A fairer comparison needs to turn these presolving methods off.
>
> A2: The table below presents the experimental results with presolving turned off for both Gurobi and our proposed method.
>
> |        | $gap_{abs}$ | PDI   |
> | ------ | ------- | ----- |
> | Gurobi | 565.61  | 76.78 |
> | PS     | 420.47  | 78.20 |
> | Ours   | 410.58  | 77.29 |
>
> It can be observed that even with presolving disabled, our method still maintains advantages over variable-based reduction approaches, demonstrating the effectiveness of constraint-based reduction.
>
> > ###  Q1. What is the setting used in the experiments for the MILP Solver?
>
> A3:  Here, we report the setting used in the experiments for the MILP Solver--Gurobi and SCIP.
>
> Gurobi setting:
> ```python
>     m.Params.TimeLimit = 800
>     m.Params.Threads = 1
>     m.Params.MIPFocus = 1
> ```
> Scip setting:
> ```
>     m1.setParam('limits/time', 800)
>     m1.setIntParam("lp/threads", 1)
>     m1.setParam('randomization/randomseedshift', 0)
>     m1.setParam('randomization/lpseed', 0)
>     m1.setParam('randomization/permutationseed', 0)
>     m1.setHeuristics(SCIP_PARAMSETTING.AGGRESSIVE) # MIPFocus
> ```
> Except for the aforementioned settings, all other parameters adopt the default configuration.

---

> > ### Comment · Reviewer_HaxL · 2025-11-27
> >
> > Thanks for the swift reply. However, my concern about this interesting paper still exists. 1) What is the dataset used in Table A2? 2) Why is the performance enhancement much smaller than the one shown in the main paper? 3) The ideal comparison is to turn off the presolving in GUROBI used in the proposed method, and the baseline GUROBI should turn the presolving on. Since the quality of the experimental part is much below my expectations. I will lower my score.

---

> > > ### Author Response · Authors · 2025-12-02
> > >
> > > We now understand your concern more clearly and apologize for the earlier lack of clarity. Below we report complete presolve ablation results on the CA and MVC datasets with Gurobi (default ```presolve=False```).  As shown, our method combined with presolve achieves the best overall performance.
> > >
> > > > **MVC based Gurobi**
> > >
> > > |        | $gap_{abs}$ | PDI   |
> > > | ------ | ------- | ----- |
> > > | Gurobi | 8.38  | 68.02 |
> > > | + presolve | 7.44  | 72.31 |
> > > |||
> > > | PS     | 0.28  | 7.39 |
> > > | + presolve | 0.20  | 8.28 |
> > > |||
> > > | Ours   | 0.00  | 6.26 |
> > > | + presolve | 0.00 | 4.56 |
> > >
> > >
> > > > **CA based Gurobi**
> > >
> > > |        | $gap_{abs}$ | PDI   |
> > > | ------ | ------- | ----- |
> > > | Gurobi | 565.61  | 76.78 |
> > > | + presolve | 477.51   | 76.29 |
> > > |||
> > > | PS     | 420.47  | 78.20 |
> > > | + presolve | 210.19  | 73.78 |
> > > |||
> > > | Ours   | 410.58  | 77.29 |
> > > | + presolve | 104.72  | 73.02 |
> > >
> > >
> > > From these results, we observe that:
> > >
> > > >1. Combining our reduction with presolve yields a further improvement over using either component alone, demonstrating their complementarity.
> > > 2. Under the same solver configuration, our learned reduction achieves better solution quality than the classical presolve;
> > > 3. For Gurobi as well as both reduction methods, enabling presolve consistently improves performance;
> > >
> > > We would like to clarify that the learned model reduction and Gurobi’s presolve are not conflicting mechanisms; they are complementary and can be used together.
> > >
> > > To better highlight the actual contribution of the learned reduction, we will incorporate the above presolve ablation results and the corresponding discussion into the main paper and appendix, so as to systematically address your concerns about the experimental setup and fairness of the comparison.

---

### Official Review · Reviewer_Ni7m · 2025-10-31

**Soundness:** 4
**Presentation:** 3
**Contribution:** 4
**Rating:** 8
**Confidence:** 5

**Summary:**

This paper presents a novel framework for accelerating Mixed Integer Linear Programming (MILP) solving by introducing the concept of Critical Tight Constraints (CTCs) — a subset of constraints that are most influential in determining the final optimal solution. The key idea is that by identifying and reducing redundant or weakly active constraints, one can substantially decrease solver time without compromising solution accuracy.

To efficiently identify CTCs, the authors design a novel framework that combines multi-dimensional MILP representation learning and constraint-level embeddings with pretrained language model (PLM) augmentation. This hybrid modeling enables the system to capture both structural and semantic relationships among constraints. Once CTCs are identified, the solver performs targeted reduction to simplify the MILP instance before solving. Extensive experiments conducted on large-scale MILP benchmarks, using Gurobi as the backend solver, demonstrate significant runtime reductions while maintaining high solution fidelity.

**Strengths:**

1. The notion of Critical Tight Constraints (CTCs) represents a fresh and impactful contribution to the MILP optimization literature. Unlike prior works focusing on branching, cut selection, or presolving heuristics, this paper introduces a new perspective centered on constraint-level importance.

2. Constraint reduction directly improves solver efficiency, making the method immediately relevant to industrial-scale applications such as logistics, scheduling, and network design.

3. The proposed neural framework integrates multiple components — structural MILP encoding, PLM-guided feature enhancement, and supervised CTC prediction — in a cohesive and well-justified pipeline.

4. The experiments are comprehensive and clearly demonstrate performance improvements over baseline solvers. The consistent gains on Gurobi are particularly convincing, showing both scalability and generalization.

5. The paper is well-written and logically organized. The motivation for focusing on constraint reduction is clearly articulated and supported by both intuition and empirical evidence.

**Weaknesses:**

1. While the intuition behind identifying CTCs is compelling, the paper would benefit from a theoretical discussion or empirical analysis showing why certain constraints consistently dominate others. Some sensitivity or ablation studies on the constraint structure could strengthen the justification.

2. The integration of PLM embeddings is an interesting choice, but it would help to quantify their actual contribution via ablation — e.g., selection of different PLM. It would be helpful to provide additional implementation details on how CTC reduction interacts with Gurobi and whether the approach is solver-agnostic.

3. The paper is quite dense, covering multiple modeling, training, and experimental components. While this reflects solid effort, the narrative could benefit from streamlining or reorganization to emphasize the core technical contributions better.

**Questions:**

1. It would be helpful to include a sensitivity analysis of the hyperparameter Δc, which controls the constraint search or reduction threshold. This experiment could clarify how robust the proposed method is across different datasets and problem scales, and whether performance degrades significantly under suboptimal Δc values.

2. The current experiments are convincing, but primarily conducted on moderate-scale benchmarks. Could the authors provide additional results or analysis on large-scale MILP instances with more complex and heterogeneous constraints? Such evaluation would strengthen the claim that the proposed reduction framework generalizes well beyond the tested benchmarks.

3. The proposed method identifies Critical Tight Constraints through a neural model. It would be interesting to see an analysis of what kinds of constraints are typically recognized as critical—e.g., are they associated with certain structural or semantic patterns? Such insights could improve the interpretability and trustworthiness of the model’s predictions.

4. Since modern MILP solvers (like Gurobi) already include powerful presolve and constraint simplification modules, could the authors clarify how their learned reduction interacts with, or complements, the built-in presolver? A discussion or ablation comparing the two would help isolate the true contribution of the learned component.

---

> ### Author Response · Authors · 2025-11-19
>
> We are sincerely grateful to the reviewer for their thoughtful, detailed, and incredibly constructive review.
>
> The reviewer's suggestions for strengthening the paper were so clear and insightful that they spurred us to conduct several new, targeted experiments, which we believe have substantially improved our work. We are pleased to present our detailed responses below, addressing each weakness and question point-by-point. We hope our responses, supported by new data, fully clarify the points raised.
> > ###  W1. While the intuition behind identifying CTCs is compelling, the paper would benefit from a theoretical discussion or empirical analysis showing why certain constraints consistently dominate others. Some sensitivity or ablation studies on the constraint structure could strengthen the justification.
>
> It can be seen from the table below that the solving performance of Best Fixing is far superior to that of Worst Fixing. In the CA problem, we observe that fixing constraints with fewer variables generally leads to faster solutions. This finding is consistent with the fact that a constraint with fewer variables has a lower Fixed Constraints Strength $\rho = \frac{n}{n+1}$.
> This observation further supports the effectiveness of our proposed definition of “critical tight constraints.”
>
> | Time (s)    | CA_easy | WA       |
> |-------------|---------|----------|
> | Original    | 378.226 | >3600    |
> | Best Fixing | 1.851   | 50.726   |
> | Worst Fixing| 465.74  | >3600    |
>
> Empirical evidence indicates critical constraints are mostly of
> 1. Set Packing ($\sum x_i \leq 1, x_i \in \{0,1\}, \forall i$ ),
> 2. Set Covering($\sum x_i \geq 1, x_i \in \{0,1\}, \forall i$),
> 3. Invariant Knapsack($\sum x_i \leq b, x_i \in \{0,1\}, \forall i, b \in \mathbb{N}_{>2}$).
>
> Motivated by the observation that different constraints exert distinct impacts, we designed our method accordingly.
>
>
>
> > ###  W2. The integration of PLM embeddings is an interesting choice, but it would help to quantify their actual contribution via ablation — e.g., selection of different PLM. It would be helpful to provide additional implementation details on how CTC reduction interacts with Gurobi and whether the approach is solver-agnostic.
>
> In this study, we initially employed the T5-base model for text encoding. To further validate the performance, we introduced the more powerful OpenAI Text-Embedding-3-Large model for comparative experiments, while keeping the intermediate feature dimension unchanged. The relevant experimental results based on Gurobi are presented in the table below (all other settings remain the same as in the original experiments).
>
> |      | T5-base | OpenAI Text-Embedding-3-Large |
> | ---- | ------- | ----------------------------- |
> | loss | 1246.3  | 1232.1                        |
> | gap  | 104.72  | 103.33                        |
> | PDI  | 73.02   | 72.96                         |
>
> As observed, the choice of PLM has limited impact. The reason is that compared with the coarse-grained sentence-level embeddings produced by PLMs, the MLP used to align PLM embeddings with GNN embeddings plays a more critical role, thereby mitigating the influence of the PLM itself.
>
> ### Furthermore, regarding your **second** question:
> it is important to emphasize that our method is solver-agnostic. An MILP instance is first reduced using our method to produce a simplified model, which is then passed to the solver for optimization. Both the reduction and search components are implemented by reconstructing the MILP instance before solver invocation, ensuring that the approach remains independent of any specific solver.
>
>
> > ###  W3. The paper is quite dense, covering multiple modeling, training, and experimental components. While this reflects solid effort, the narrative could benefit from streamlining or reorganization to emphasize the core technical contributions better.
>
> Thank you for your affirmations and comments. Our core contribution lies in proposing the first constraint-based general model reduction method, which we will reorganize and emphasize in the introduction.
> Below is the reorganized introduction:
>
> Our **first** contribution is the proposal of a constraint-based general model reduction framework. We design a heuristic algorithm to identify and learn critical tight constraints, and introduce a constraint search neighborhood to mitigate inaccuracies arising from neural network predictions.
>
> The **second** contribution is a novel multi-modal representation that incorporates abstract models into instance models, thereby enhancing the predictive capability of neural networks.
>
> **Finally**, we conduct extensive experiments on large-scale MILP domains to validate the effectiveness of the proposed constraint-based model reduction method.

---

> > ### Author Response · Authors · 2025-11-19
> >
> > > ###  Q1. It would be helpful to include a sensitivity analysis of the hyperparameter Δc, which controls the constraint search or reduction threshold. This experiment could clarify how robust the proposed method is across different datasets and problem scales, and whether performance degrades significantly under suboptimal Δc values.
> >
> > The table below shows the sensitivity of our method to the hyper-parameters $k_c$ and $\Delta_c$ (sensitivity results for variable-related hyper-parameters are provided in other variable-based work).
> >
> > |               | Gurobi  |       | SCIP    |        |
> > | ------------- | ------- | ----- | ------- | ------ |
> > |               | $gap_{abs}$ | PDI   | $gap_{abs}$| PDI    |
> > | **(10, 0)**   | 492.78  | 74.88 | 3945.76 | 111.03 |
> > | **(20, 0)**   | 109.57  | 73.07 | 3401.63 | 109.41 |
> > | **(40, 5)**   | 375.73  | 73.56 | 4270.87 | 114.64 |
> > | **(50, 5)**   | 228.69  | 73.52 | 4054.70 | 108.69 |
> > | **(80, 15)**  | 104.72  | 73.02 | 3485.82 | 108.23 |
> > | **(100, 20)** | 161.81  | 73.41 | 3673.37 | 109.11 |
> > | **(120, 25)** | 217.94  | 73.98 | 4535.83 | 110.73 |
> >
> >
> > > ###  Q2. The current experiments are convincing, but primarily conducted on moderate-scale benchmarks. Could the authors provide additional results or analysis on large-scale MILP instances with more complex and heterogeneous constraints? Such evaluation would strengthen the claim that the proposed reduction framework generalizes well beyond the tested benchmarks.
> >
> > In the table below, we report the results for the SCUC problem under the extremely large-scale case2868rte scenario, which features highly complex constraint structures and involves an average of over 600,000 variables and constraints. We trained the model on case118 (a scenario already extremely challenging due to its large scale) and then generalized the trained model to the case2868rte scenario.
> >
> > |method|$gap_{abs}$|PDI|
> > |--|--|--|
> > |PS|2953.56|9.98|
> > |**ours**|**2927.53**|**9.80**|
> >
> > We also present the result for one instance from MIPLIB—*highschool1-aigio*, which is a particularly challenging case. We solve this hard instance based on the solver OPTV.
> >
> > |method | Time |
> > |-|-|
> > |OPTV | 7200s |
> > |Ours | 682s|
> >
> >
> >
> > > ###  Q3. The proposed method identifies Critical Tight Constraints through a neural model. It would be interesting to see an analysis of what kinds of constraints are typically recognized as critical—e.g., are they associated with certain structural or semantic patterns? Such insights could improve the interpretability and trustworthiness of the model’s predictions.
> >
> > Our critical tight constraints are defined based on heuristic rules. We observed significant variations in solution time when solving with randomly fixed constraints, leading to the finding that critical constraints are predominantly of the
> > 1. Set Packing ($\sum x_i \leq 1,; x_i \in {0,1},; \forall i$),
> > 2. Set Covering ($\sum x_i \geq 1,; x_i \in {0,1},; \forall i$),
> > 3. Invariant Knapsack ($\sum x_i \leq b,; x_i \in {0,1},; \forall i,; b \in \mathbb{N}_{>2}$)
> >
> > categories—an insight substantiated by our experimental results.
> >
> > Theoretically, we prove that fixing such constraints yields greater information gain, and empirical results further demonstrate the effectiveness of our method.
> >
> >
> >
> >
> > > ###  Q4. Since modern MILP solvers (like Gurobi) already include powerful presolve and constraint simplification modules, could the authors clarify how their learned reduction interacts with, or complements, the built-in presolver? A discussion or ablation comparing the two would help isolate the true contribution of the learned component.
> >
> > The table below presents the experimental results with presolving turned off for both Gurobi and our proposed method.
> >
> > |        | $gap_{abs}$ | PDI   |
> > | ------ | ------- | ----- |
> > | Gurobi | 565.61  | 76.78 |
> > | PS     | 420.47  | 78.20 |
> > | Ours   | 410.58  | 77.29 |
> >
> > It can be observed that even with presolving disabled, our method continues to outperform variable-based reduction approaches, demonstrating the effectiveness of constraint-based reduction.

---

> > > ### Comment · Reviewer_Ni7m · 2025-11-24
> > >
> > > Thank you for your detailed rebuttal, which addressed my concerns regarding sensitivity and large-scale performance. I will maintain my original score of 8, as the paper presents a solid contribution, and I look forward to the final version.

---

### Official Review · Reviewer_ajfo · 2025-11-01

**Soundness:** 3
**Presentation:** 3
**Contribution:** 3
**Rating:** 6
**Confidence:** 3

**Summary:**

This paper proposes predicting tight constraints to perform constraint reduction in MILP problems, effectively accelerating MILP solving. To better predict tight constraints, in addition to the existing bipartite graph representation, the paper introduces an abstract-level representation of MILP and a TCP module.

**Strengths:**

1. Every component proposed or used in the paper is well-motivated.
2. The definition and use of the fixed constraint strength $\rho$ are quite interesting.

**Weaknesses:**

1. Introducing a pretrained language model and the abstract-level GNN will bring additional computational overhead.
2. I have doubts about Theorem 4. If an error occurs during the process of fixing constraints, would the resulting problem become infeasible, or would its optimal value differ from that of the original problem?
3. In Table 8, I noticed that the hyperparameters used for different problem types vary significantly. This implies that for a new type of problem, the proposed method would require manual hyperparameter tuning. I would like to know how sensitive the method’s performance is to these hyperparameters.

**Questions:**

1. When the lower or upper bounds of constraints in MILP problems are no longer 0 or 1, how does the proposed constraint reduction method handle such cases, and how does this affect performance in practice?
2. Could the paper provide experimental results on more complex real-world problems (such as those from MIPLIB)? For a real-world problem of an unknown type, how is its text description obtained?
3. How significant is the performance impact when using different pretrained language models?

---

> ### Author Response · Authors · 2025-11-19
>
> We are sincerely grateful to the reviewer for their thoughtful, detailed, and incredibly constructive review.
>
> The reviewer's suggestions for strengthening the paper were so clear and insightful that they spurred us to conduct several new, targeted experiments, which we believe have substantially improved our work. We are pleased to present our detailed responses below, addressing each weakness and question point-by-point. We hope our responses, supported by new data, fully clarify the points raised.
>
> > ###  W1. Introducing a pretrained language model and the abstract-level GNN will bring additional computational overhead.
>
> A1: Your point is well-taken. We would like to clarify that the additional computation introduced by our method appears only during the training phase and does **not** affect the solution time in the online phase, which is typically the primary concern. As shown in the table below, the online inference time remains essentially unchanged.
>
> |          | Inference Time |
> | -------- | -------------- |
> | PS (GNN) | 337.72 ms      |
> | Ours     | 346.24 ms      |
>
>
>
> > ### W2. I have doubts about Theorem 4. If an error occurs during the process of fixing constraints, would the resulting problem become infeasible, or would its optimal value differ from that of the original problem?
>
>
> A2: It should be clarified that Theorem 4 states that if the neural network’s prediction is correct, both optimality and feasibility are guaranteed. However, if the prediction is inaccurate, suboptimality may occur. To address this, we introduce a constraint search mechanism (Formula 12) to mitigate the impact of such prediction errors.
>
> > ### W3. In Table 8, I noticed that the hyperparameters used for different problem types vary significantly. This implies that for a new type of problem, the proposed method would require manual hyperparameter tuning. I would like to know how sensitive the method’s performance is to these hyperparameters.
>
> A3: To clarify, the variation in hyper-parameter values is driven by differences in problem scale and follows a principled tuning procedure. In practice, we tune them by inspection on the training set.
> The table below shows the sensitivity of our method to the hyperparameters $k_c$ and $\Delta_c$ (sensitivity results for variable-related hyperparameters are provided in other variable-based works).
>
> |               | Gurobi  |       | SCIP    |        |
> | ------------- | ------- | ----- | ------- | ------ |
> |               |$gap_{abs}$ | PDI   | $gap_{abs}$ | PDI    |
> | **(10, 0)**   | 492.78  | 74.88 | 3945.76 | 111.03 |
> | **(20, 0)**   | 109.57  | 73.07 | 3401.63 | 109.41 |
> | **(40, 5)**   | 375.73  | 73.56 | 4270.87 | 114.64 |
> | **(50, 5)**   | 228.69  | 73.52 | 4054.70 | 108.69 |
> | **(80, 15)**  | 104.72  | 73.02 | 3485.82 | 108.23 |
> | **(100, 20)** | 161.81  | 73.41 | 3673.37 | 109.11 |
> | **(120, 25)** | 217.94  | 73.98 | 4535.83 | 110.73 |
>
> When generalizing to problems of the same type but different sizes, the hyperparameters can be set proportionally. Below we present the hyperparameters used for two scales of CA.
>
> | scale      | $k_0$ | $k_1$ | $\Delta$ | $k_c$ | $\Delta_c$ |
> | ---------- | ----- | ----- | -------- | ----- | ---------- |
> | (300,1500) | 1000  | 0     | 10       | 80    | 15         |
> | (600,3000) | 1200  | 0     | 12       | 100   | 10         |
>
> The hyperparameters for the two scales differ by roughly a factor of 1.2.
>
> > ###  Q1. When the lower or upper bounds of constraints in MILP problems are no longer 0 or 1, how does the proposed constraint reduction method handle such cases, and how does this affect performance in practice?
>
> A4: Thank you for the question. In fact, our method is not restricted to 0/1 bounds; for any other constraint type, we can likewise pre-compute $\rho$ and perform the same comparison. For instance, the constraints in the WA dataset have non-{0,1} bounds.
> Here are the main constraints of WA:
>
> 1. **Activation-Capacity Association Constraint**：$y_{ij} \leq C_i x_i, \quad \forall i \in I, \forall j \in J$,
> 2. **Worker Total Capacity Constraint**：$\sum_{j \in J} y_{ij} \leq C_i, \quad \forall i \in I$,
> 3. **Single-Failure Robustness Constraint**：$\sum_{k \in I \setminus \{i\}} y_{kj} \geq d_j, \quad \forall j \in J, \forall i \in I$.
>
> Both the Worker Total Capacity Constraint and the Single-Failure Robustness Constraint can be seen to have non-{0,1} bounds. And our method also demonstrates strong performance on the WA dataset.
>
> ||$gap_{abs}$|PDI|
> |-|-|-|
> |Gurobi| 0.20 | 4.76 |
> |PS| 0.07 |  4.97 |
> |**Ours**| **0.06** |  **4.40** |

---

> ### Author Response · Authors · 2025-11-19
>
> > ###  Q2. Could the paper provide experimental results on more complex real-world problems (such as those from MIPLIB)? For a real-world problem of an unknown type, how is its text description obtained?
>
>
> A4: In the table below, we report the results for the SCUC problem under the extremely large-scale case2868rte scenario, which features highly complex constraint structures and involves an average of over 600,000 variables and constraints. We trained the model on case118 (a scenario already extremely challenging due to its large scale) and then generalized the trained model to the case2868rte scenario.
>
> |method|$gap_{abs}$|PDI|
> |--|--|--|
> |PS|2953.56|9.98|
> |**ours**|**2927.53**|**9.80**|
>
> We also present the result for one instance from MIPLIB—*highschool1-aigio*, which is a particularly challenging case. We solve this hard instance using the solver OPTV with a 7200s time limit.
>
> |method | Time |
> |-|-|
> |OPTV | 7200s |
> |Ours | 682s|
>
> ###  Regarding your second question about the text description of unknown-type problems:
> such text descriptions are common and readily available for many practical MILP tasks (e.g., supply chain management and power dispatch). Real-world problems typically originate from concrete scenarios that experts then abstract into a MILP formulation.
> Even in the extreme case where no text description is available, our constraint reduction method remains effective, as demonstrated by the ablation results in the table below.
>
> |method|$gap_{abs}$|PDI|
> |-|-|-|
> |Gurobi|477.51|76.29|
> | PS |210.19|73.78|
> |ours + w/o text |118.27|73.36|
> | ours| 104.72 | 73.02|
>
>
> > ###  Q3. How significant is the performance impact when using different pretrained language models?
>
> A5: In this study, we initially employed the T5-base model for text encoding. To further validate the performance, we introduced the more powerful OpenAI Text-Embedding-3-Large model for comparative experiments, while keeping the intermediate feature dimension unchanged. The relevant experimental results based on Gurobi are presented in the table below (all other settings remain the same as in the original experiments).
>
> |      | T5-base | OpenAI Text-Embedding-3-Large |
> | ---- | ------- | ----------------------------- |
> | loss | 1246.3  | 1232.1                        |
> | gap  | 104.72  | 103.33                        |
> | PDI  | 73.02   | 72.96                         |
>
> As observed, the choice of PLM has a limited impact. The reason is that compared with the coarse-grained sentence-level embeddings produced by PLMs, the MLP used to align PLM embeddings with GNN embeddings plays a more critical role, thereby mitigating the influence of the PLM itself.
>
> -----
> Once again, we wish to express our profound gratitude for your detailed and constructive review. Your insightful feedback has been instrumental in helping us to refine our arguments and better showcase the full scope and impact of our work.
>
> We have sought to address each of your points thoroughly and hope our responses and new experimental results have clarified the contributions of our paper. We eagerly welcome any further questions or discussion and look forward to the opportunity to continue improving our work based on your expertise.

---

### Author Response · Authors · 2025-12-02
**Summary of Review and Discussion**

First, we thank all reviewers for their insightful feedback. We are thrilled they found our work is well-motivated (```ajfo, 3DQK```) and proposes an interesting and impact method(```ajfo, HaxL, 3DQK```),  our manuscript is praised as "well-written with a clear and logical flow" (```3DQK,Ni7m```). Both the 1st and 2nd reviewers ```ajfo (6)``` and ```Ni7m (8)``` recommended acceptance from the start.

----

Our rebuttal try our best to addressed the reviewers' initial concerns, providing comprehensive and thorough new analysis (```Ni7m, HaxL```) and clarifying our methodology and key contributions ( ```Ni7m, 3DQK, ajfo``` ).  The 4th reviewer `3DQK` explicitly stated that their main concern had been resolved and increased their score from **4** to **6**, as recorded in the discussion history.  We also addressed reviewer `Ni7m`’s concerns, and they retained their original score of **8**. In addition, we were engaged in a deeper discussion with 3rd reviewer `HaxL` regarding the ablation study on presolve.

### **1. core concern A: The ablation study related to presolve**
#### (Raised by Reviewers `Ni7m` [Question 4], `HaxL` [Weakness 2])

**Our response**:  We report complete presolve ablation results on the CA and MVC datasets with Gurobi (default ```presolve=False```). Under the same solver configuration, **our learned reduction achieves better solution quality than the classical presolve and our method combined with presolve achieves the best overall performance**.

| Method         | MVC $gap_{abs}$ | MVC PDI | CA $gap_{abs}$ | CA PDI |
|----------------|-----------------|---------|----------------|--------|
| Gurobi         | 8.38    | 68.02   | 565.61         | 76.78  |
| + presolve     | 7.44    | 72.31   | 477.51         | 76.29  |
| PS             | 0.28    | 7.39    | 420.47         | 78.20  |
| + presolve     | 0.20    | 8.28    | 210.19         | 73.78  |
| Ours           | 0.00    | 6.26    | 410.58         | 77.29  |
| + presolve     | 0.00    | 4.56    | 104.72         | 73.02  |

**Conclusion**: By performing comprehensive ablation studies on presolve, we were able to better highlight the actual contribution of the learned reduction and demonstrate the effectiveness of our method. Our response resolved the concerns of `Ni7m` on this point, and we are confident that it fully addresses the related concerns raised by `HaxL`.

### **2. core concern B: The local decoupling assumption is quite strong and its applicability in practice remains unclear.**
#### (Raised by Reviewers `HaxL` [Weakness 1], `3DQK` [Weakness 2], **which has been addressed**)

**Our response**: For Assumption 1, we intend to show that the method admits favorable theoretical properties when the MILP exhibits certain structural characteristics. Without these characteristics, our proposed method remains effective.
We further provide statistical evidence to quantify the gap between theory and practice, and the results demonstrate a strong consistency between our theoretical findings and empirical observations.

**Conclusion**: We introduce the assumption purely as an analytic tool; our method itself does not rely on it. Our response resolved the concerns of `3DQK`, and they raised their score from **4 to 6** as evidenced by the discussion history.

----
Based on all reviewers’ comments and suggestions, we have revised our manuscript and updated the submission. The main additional contents are as follows:

1. Tight Constraints and Fixing. (Clarification of the definition of "fixing") [section 3]
2. Definition 1 (Further clarification of "$C_i(x)$ is satisfied") [section 4.2]
3. proposition 3 (To avoid potential misunderstandings regarding Assumption) [section 4.2]
4. Table 12 (Presolve ablation study) [Appendix H.4]
5. Table 14 (Effect of the pretrained language model on our method) [Appendix H.5]
6. Table 16 (Empirical validation of the local decoupling assumption on MILP benchmarks) [Appendix H.7]

---
Our method has been consistently recognized as well-motivated and impactful, and we believe that our rebuttal has addressed the reviewers’ remaining concerns.
The entire review process has been invaluable in strengthening the paper, and we are confident in its contribution to the ICLR community.

---

### Meta-Review · Area_Chair_vyPP · 2026-01-08

**Summary:**

The paper proposes a novel constraint-based model reduction framework for Mixed-Integer Linear Programming (MILP) that utilizes a multi-modal representation to identify and fix critical tight constraints. The reviewers and authors were actively involved in the rebuttal process. In response to the main concerns, the authors made substantial efforts and improved the quality of the paper.

Overall, most concerns have been addressed to a satisfactory degree, and the submission appears to offer a meaningful contribution to ML-for-optimization. I therefore believe the paper is worthy of publication. For the camera-ready version, I encourage the authors to thoroughly review the meta-review and all reviewer comments to ensure that all promised revisions and new results are fully incorporated and clearly documented.

**Reviewer Concerns:**

### [ajfo]

W2: Regarding Theorem 4, I agree with the reviewers that while the core logic is substantially correct, the current statement is too informal for a technical theorem. Specifically, terms like "before and after fixing" are colloquial and lack mathematical precision. A formal theorem requires rigorous definitions of the feasible regions $\mathcal{F}$ and $\mathcal{F}'$, along with an explicit hypothesis that the subset of constraints being converted to equalities corresponds strictly to constraints that are active (tight) at the optimal solution $x^*$. Without these precise conditions, the theorem statement is technically incomplete.

However, since theory is not the primary focus of this work and the proof itself relies on a straightforward property of tight constraints, I suggest removing the formal "Theorem" environment to avoid overclaiming. Instead, please consider one of the following revisions:(i) Replace the theorem with an informal justification in the main text describing why "fixing partial tight constraints still maintains the feasibility and optimality as the original problem" (provided the predictions are correct). (ii) Move the logic from Appendix B.2 into the main text and present it as a derivation or property. Since the proof is trivial and easy to follow, it serves better as an explanatory step in your methodology rather than a standalone theorem.

Other concerns have been addressed.

### [Ni7m]

All concerns are addressed.

### [HaxL]

All concerns are finally addressed. Although the reviewer initially questioned the experiments during the first round of rebuttal, the systematic results provided in the final response have substantially resolved these doubts.

### [3DQK]

All concerns are addressed.

**Reviewer Scores:**

Reviewer ajfo is likely to keep a score of 6, as most of their concerns are addressed.

Ni7m will keep the score of 8 because all their concerns are addressed.

HaxL is expected to maintain a score of 4 or potentially raise it to 6. While they expressed significant doubts regarding the experiments during the first round of rebuttal, the subsequent systematic ablation studies appear to have substantially resolved these concerns.

3DQK will raise the score from 4 to 6.

---

### Decision · Program_Chairs · 2026-01-26

Accept (Poster)